# Unraveling the Dynamics of Stable and Curious Audiences in Web Systems

## ABSTRACT

In this paper, we propose **BPoP** (**Burst-induced Poisson Process**), a parsimonious model to analyze time series, such as Twitter feeds or Youtube search queries. **BPoP** is able to disentangle the slowly-varying regular activity of the *stable audience* from the *curious audience* activity occurring in bursts such as viral threads. Our model is a mixture two hidden and interacting processes. The first component is a self-feeding-process (SFP), and the second is a stochastically driven Poisson process with a random step function as intensity function, whose transitions are caused by the bursty behavior of the first component. The SFP generates a bursty behavior, corresponding to viral threads caused by sudden external events, whereas the non-homogeneous Poisson process models normal background behavior that is influenced only by the overall popularity of the topic (the *stable audience*). We performed extensive empirical work that shows that our model fits and characterizes a large number of real datasets with better results than state-of-art models. More importantly, we show that **BPoP** is able to quantify the *stable audience* of media channels over time, which may serve as a good indicator for their popularity.

**ACM Reference Format:**
Anonymous Author(s). 2023. Unraveling the Dynamics of Stable and Curious Audiences in Web Systems . In *Proceedings of ACM Conference (Conference'17)*. ACM, New York, NY, USA, 16 pages. https://doi.org/10.1145/nnnnnnn.nnnnnnn

## 1 INTRODUCTION

Why do content creators on YouTube frequently request their subscribers to enable all notifications? This practice stems from their awareness that not all subscribers are regular viewers of their channels [1], despite the argument that the primary determinant of sustained interest lies in the number of subscriptions a channel garners [2, 3]. In fact, the popularity dynamics of online items can be explained by several endogenous and exogenous factors, which include the quality of the content [4, 5], their metadata [6], their age [7], the recommendation algorithm and its rank on keyword-based queries [8], promotions [9], and social network effects [4, 10].

Accurately predicting the enduring appeal of online content remains an exceptionally challenging task due to the distinct patterns exhibited in the popularity dynamics of online items [11–13]. These dynamics often involve one or more peaks of *popularity bursts* that

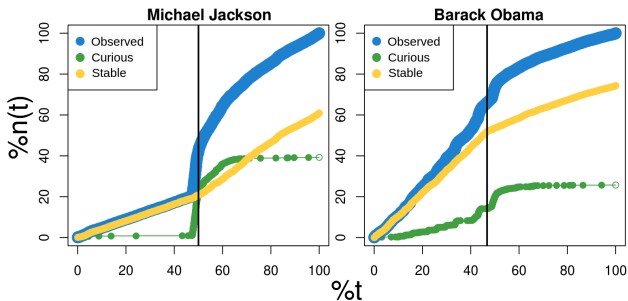

**Figure 1: Two example time series that motivate the proposed time-series model. Left: Michael Jackson (Jan/01/2008 to Dec/31/2010). Right: Barack Obama (Jul 20 2015 Jul 19 2018). The data is taken from Google Trend (country: USA; search engine: Youtube). A description of the results is given in the main text below.**

intermingle with the *regular and stable audience* of the content. Existing literature predominantly focuses on supervised and feature-based approaches for predicting long-term popularity [5, 9, 13–15], while some models overlook the bursty nature of popularity dynamics [16–18]. Moreover, these approaches aim to forecast an item's overall popularity, taking into account the influence of exogenous factors such as media exposure and virality on social media.

Unlike previous studies, this paper introduces a novel approach that (1) distinguishes between *popularity bursts* and the *consistent and stable audience* of content, and (2) investigates the underlying dynamics of these audiences. We specifically focus on differentiating two audience types: the *curious* audience, attracted by external and viral events such as gossip [9, 19, 20], and the *stable* audience, representing stable viewership. Empirical evidence reveals that content such as keyword-discovered videos [7], popular TV episodes, and music videos [4] maintains steady popularity over time, dominated by the *stable* audience. On the other hand, news, sports, and movie content often undergo rapid popularity surges followed by quick declines, mostly due to temporally limited events (e.g. breaking news). In these cases, the *curious* audience tends to prevail over the *stable* audience.

The primary challenge in distinguishing between these audience types arises from the lack of individual labels that distinguish *stable* and *curious* viewers. Instead, we typically only have the total number of viewers for an observed random series of events (RSE), which is a combination of both hidden processes associated with the two types of audiences. Disentangling these two audience types poses difficulties, particularly because during viral events, *curious* users tend to dominate the channel's activity, leading to a burst in the overall RSE [21–23]. Such bursts can become so prominent that they completely obscure the presence of the *stable audience* during these events. To accurately identify and quantify the *stable audience*, it

becomes essential to pinpoint the timestamps associated with these bursts. However, this task is especially challenging as the activity from the *stable audience* remains unlabelled and gets mixed with the bursts.

Another significant challenge arises from the potential for the *stable audience* to modify its typical behavior in response to bursts or external events [24–30]. For instance, the unexpected death of Michael Jackson in June 2009 triggered a surge in media and web activity, leading to increased music sales, video views, and reactions in posts and comments. During this period, both existing and new fans engaged with Jackson's work, transforming him into an enduring musical icon. This behavior is illustrated in the left-hand side of Figure 1, where the blue line represents the cumulative web activity associated with Michael Jackson's RSE. Initially, it displayed a relatively constant growth rate until his death (vertical black line), followed by a sudden spike in interest. Over time, the blue line returned to a consistent growth rate. We will illustrate how to distinguish between two types of web activities during such events: (1) regular *stable audience* activity (yellow line) and (2) activity driven by unexpected events generated by the *curious* audience (green line). Notably, the yellow curve changed its slope after Jackson's unexpected death, marking a significant transition event that not only led to a short-term burst of activity but also *permanently altered* the *stable audience* [27, 30]. The revival of his songs, tribute notes, and the younger generation's discovery of Jackson's work contributed to a sustained increase in web interaction. Conversely, the end of Barack Obama's presidential term (right-hand side of Figure 1) resulted in reduced political activity and a decline in mentions.

Point processes form a statistical framework to learn and infer about RSEs [31, 32]. There are two contrasting approaches in this domain. One focuses on self-exciting point processes, which model correlations between past and future events [19, 33, 34]. On the other hand, the homogeneous Poisson process and its variants have also been deemed suitable [19, 35–37]. This divergence has led to extensive research examining the diversity of human actions. For example, studies have found that Twitter hashtag activity can be continuous, periodic, or concentrated around a single peak [38]. Similarly, research on YouTube videos has shown that the current tweeting rate and tweet volume since a video's upload are crucial parameters for identifying its virality or popularity [39]. Additionally, the popularity of YouTube videos can undergo multiple phases of growth and decline, likely influenced by various background random processes superimposed on bursty behavior [24].

Therefore, in theory, point processes could be used to solve the problem of estimating the *stable audience* of online items, but existing models are not appropriate for this particular setting: they focus on different aspects of RSE characterization, and do not provide methods to identify and measure burst-induced **changes** in *background* popularity. While Poisson processes (PPs) [36, 37] can easily estimate the *stable audience* when **all** incoming events arrive at a fixed and predictable rate, they fail to mimic the bursts of events seen in real data. On the other hand, self-exciting processes, such as Hawkes and Wold processes, are able to capture the correlations between consecutive events that generate bursts of activity, but existing approaches do not model the *time-varying* nature of the *stable audience* [18, 20, 30, 40–43].

To address these concerns, we propose the **Burst-induced Poisson Process** (**BPoP**) model, which is able to flexibly incorporate dependencies between the two hidden and underlying point processes involving the *stable audience* and the *curious*. We show that **BPoP** mimics the bursts of events seen in real data and is also able to efficiently capture the time-varying background rates that realistically represent the *stable audience*. Our main contributions are: **(a) A New Model**, namely **BPoP**, which is able to disentangle the slowly-varying regular activity of the *stable audience* from the *curious* activity occurring in bursts. This model does not depend on hard-to-get external information but uses only random series of events (RSEs) (Section 2); **(b) An EM algorithm** to cope with our intensity function's complex dependence on the history of the process (Section 3); **(c) Novel findings** describing and quantifying the *stable audience* for eleven real world data containing more than a hundred thousand RSEs (Section 4). **(d)** Extensive empirical investigations have demonstrated that **BPoP** consistently outperforms alternative models in fitting both real and synthetic data (Section 5).

## 2 THE BPOP MODEL

**Formal construction:** filtrations are formal constructions in probability theory required for the formal description of time-dependent processes. Consider a general continuous-time Markov process adapted to the filtration $(\mathcal{H}_t)_{t \in \mathbb{R}^+}$: $\mathcal{H}_t$ represents the information that is realised at time $t$. Let $N(a, b)$ be the random number of events in $(a, b)$. The conditional intensity rate function characterizes the distribution and is given by $\lambda(t|\mathcal{H}_t) = \lim_{h \to 0} \mathbb{E}\left(N(t, t+h)|\mathcal{H}_t\right)/h$.

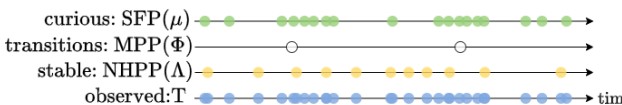

**Figure 2: The Burst-induced Poisson Process (BPoP) model. The *curious* (SFP) and *stable audience* (NHPP) labels, as well as the *transitions* (events of MPP), are not observed.**

Figure 2 shows the main idea of **BPoP**. We observe the point process timestamps $0 < t_1 < t_2 < \ldots$ of events up to a time $t$ (depicted as blue dots in the fourth row). We assume that these events are a mixture of events coming from the *stable audience* and the *curious* audience, which are two dependent point processes, represented as yellow and green dots in the first and third rows, respectively. On one hand, we model the *curious* audience generating the occasional bursts as a *Self-Feeding process* (SFP) [18, 33, 43]. SFPs are simple self-exciting processes, and their intensity function is given by $\lambda_s(t|\mathcal{H}_t) = 1/(\mu/e + \Delta t_i)$ (where $\Delta t_i = t_i - t_{i-1}$, $t_i = \max_k\{t_k : t_k \leq t\}$ and $\mu > 0$), exhibiting strong bursty behavior. On the other hand, the events associated with the *stable audience* are modelled by a second process, the classical non-homogeneous Poisson process (NHPP), shown in the third row. A third underlying meta Poisson process (MPP) controls the times when the *stable audience* (or NHPP) *transitions* occur. These *transitions* are shown as white dots in the second row in Figure 2. The intensity of the meta Poisson process generating the transitions is proportional to the

intensity of the SFP process, which acts as a soft proxy for "whether a burst is currently occurring".

**Remark:** The main difficulty with this model is that we only observe the blue dots in the fourth row. The labels associated with each event (the green and yellow colors) and the *transitions* (the white dots) are not directly observed.

Therefore, our **BPoP** model involves the combination of an SFP, representing the *curious* audience, and a NHPP, representing the *stable audience*, which interact with each other. At time $t$, the history of the process is composed of the observed event timestamps $\{t_1, t_2, \ldots\} < t$, unobserved labels $\{z_1, z_2, \ldots\}$ as well as unobserved MPP events $\{\varphi_1, \varphi_2, \ldots\}$, which represent the *transitions*. We use the convention that $z_i = 0$ if $t_i \in$ NHPP and $z_i = 1$ if $t_i \in$ SFP. Thus, **BPoP** is governed by the following three intensity functions: (1) the SFP intensity $\lambda_s(t) = 1/[(g(t) - g(g(t))) + \mu/e]$ where $g(u) = [\max(t_i : z_i = 1 \quad \wedge \quad t_i < u)]_+$ denotes the last SFP event before $t$, with the convention that $g(t) = 0$ if $\nexists i : t_i \le t \wedge z_i = 1$ and $\mu > 0$ is the SFP parameter; (2) $\lambda_\Phi(t) = c\lambda_s(t)$, where $c \in [0, 1]$ is a parameter that controls the NHPP transition sensitivity; and (3) $\lambda_p(t) = \lambda_{m_t}$, where $\Lambda = \{\lambda_0, \lambda_1, \lambda_2, \cdots\}$ is an infinite set of positive numbers (parameters) and, $m_t = \sum_{j=1}^{n_\phi} 1_{\varphi_j < t}$. Similarly we can define $\lambda_s^+(t) = \lim_{\delta \to 0} \lambda_s(t + \delta)$ (resp. $\lambda_\phi^+(t)$ and $\lambda_p^+(t)$) as the intensity of the SFP (resp. MPP and NHPP) immediately after $t$. Thus, to generate the next time stamp, we first generate three exponential variables $E_s$, $E_\phi$ and $E_p$ with intensities $\lambda_s^+(t)$, $\lambda_\phi^+(t)$ and $\lambda_p^+(t)$ respectively. Then, the next event will take place at $t + E$, where E=$\min(E_s, E_\phi, E_p)$, and it will belong to the SFP (resp. NPHH, MPP) component if $E = E_s$ (resp. $E_\phi$, $E_p$). Likewise, we continue generating the rest of the process from time $t + E$.

Considering the described generative model we aim to infer the parameters of **BPoP**. When performing inference, $c$ is set as a hyperparameter[1] and the parameters $\Lambda$, $\mu$ are determined via maximum likelihood. To optimize the likelihood, we will use the EM algorithm, relying on Gibbs sampling in the E-step. However, the EM algorithm in the case of point processes requires great care, since the events are not independent data and the usual derivations are not appropriate.

## 3 FITTING USING EM

Our optimization approach to fitting the model relies on the EM algorithm [44, 45]. The EM algorithm represents a broad class of alternating optimization methods used to estimate the maximum likelihood estimate of parameters $\theta$ in statistical models involving unobserved latent variables $Z$. The strategy consists of two steps: (1) the E-step estimates the conditional distribution of the latent variables (given the observations) based on the current estimate of the parameters; (2) the M-step computes the maximum likelihood estimate of the parameters based on the current estimate of the latent distribution. These two steps are performed alternately until convergence. In our specific case, the latent variables are the labels $z_i$ (SFP/curious versus Poisson/stable) of the observed timestamps and the transitions $m_i$, while the parameters are the $\lambda$'s and the $\mu$. In this context, since the conditional distribution of the labels (given

the observations and the parameters) is not analytically tractable, we estimate it using Gibbs sampling [46].

In this section, we compute the likelihood for our model, the marginal conditional probabilities required for Gibbs simulation, and present the overall details of our approach. To do that, we must first introduce some notation. Let $T = \{t_1, t_2, \cdots, t_n\}$ be the observed event timestamps from the mixture of the SFP (*curious*) and the NHPP (*stable audience*). Also, let $N(t) = \sum_{i=1}^{n} 1_{t_i \le t}$ be a function that computes the cumulative number of events up to time $t$. The number of *transitions* that occurred before $t_i$ is given by $M := \{m_1, m_2, \cdots, m_n\}$, where $m_i = \sum_{j=1}^{n_\phi} 1_{\varphi_j < t_i}$, and the set of NHPP transitions between $t_{i-1}$ and $t_{i+1}$ is given by $\Phi_i := \{\varphi_j | t_{i-1} < \varphi_j < t_{i+1}\}$. Finally, we set $\theta = Z \cup \Phi \cup M$ (the latent variables), $\theta_{-i} = \theta \setminus \theta_i$ where $\theta_i := (\{z_i, m_i\} \cup \Phi_i)$. Please consult the Appendix for a table of notation.

**E-Step:** Here we will explain how to use *Gibbs sampling* to draw a set of latent variables $Z$, $M$ and $\Phi$ (collectively referred to as $\theta$) from the conditional distribution given a fixed set of parameters $\mu$ and $\Lambda$. *Gibbs sampling* is a general statistical method which allows one to draw samples from complicated high-dimensional distributions. To draw a sample from a distribution $p$ on $\mathbb{R}^d$, we start with an arbitrary vector $x \in \mathbb{R}^d$, and proceed to iteratively replace each coordinate $x_i$ ($i \le d$) by a sample from the *conditional distribution* of $x_i$ given the current values of the other coordinates $x_j$ ($j \ne i$). In many situations, the conditional distribution is easier to compute than the multivariate probability density function (PDF) due to the intractability of the calculation of the multivariate normalization constant. It is known that under mild conditions, *after convergence*, Gibbs sampling leads to a sample from the original multivariate distribution $p$ [46]. In our model, the distribution to estimate is the latent *joint* distribution of the labels $z_i$ and transitions $m_i$. The distribution is proportional to the corresponding likelihood, but the normalization constant is intractable. On the other hand, the conditional marginal distributions are easy to compute, making Gibbs sampling a practical solution.

We start with an initialized value for $\theta$, and we perform a large number $N_{\text{Gibbs}}$ of updates on its components. At each update step, we pick $i \le n$ and update the value of the component $\theta_i$ according to the conditional distribution of $\theta_i$ given the current value of $\theta_{-i}$ (and, as always, the value of $T$). After a large number of iterations, this procedure yields a sample whose distribution is approximately that of a sample of $\theta$ given $T$ only. Indeed, the distribution in question is the only stationary distribution of the Markov chain corresponding to the updates, as long as the chain is aperiodic and irreducible[2]. To perform this procedure, we need to compute the conditional probability $\mathbb{P}(\theta_i | T, \theta_{-i})$ for any $i, \theta_i, \theta_{-i}$.

We will do that in two steps: first, we compute the conditional probability $\mathbb{P}(z_i, m_i | T, \theta_{-i})$, and second, we compute the conditional probability density function of $\Phi_i$ given $\theta_{-i}, T$ and $m_i, z_i$. Those conditional probabilities and densities are proportional to the corresponding likelihoods. Note that we have the following expression for the likelihood of our model $\mathcal{L}(\theta) =$

$$\prod_{i=1}^{n} \lambda_s(t_i)^{z_i} \lambda_p(t_i)^{1-z_i} \prod_{j=1}^{n_\phi} \lambda_\phi(\varphi_j) \times e^{-\int_0^{t_n} \lambda_s(t) + \lambda_\phi(t) + \lambda_p(t)dt}.$$
$$(1)$$

---

[1]Indeed, one cannot simply optimize over it since larger $c$ allows for far more transitions and makes the model prone to overfitting

[2]Those properties follow from the fact that the conditional distributions considered all have full support, as can be seen below.

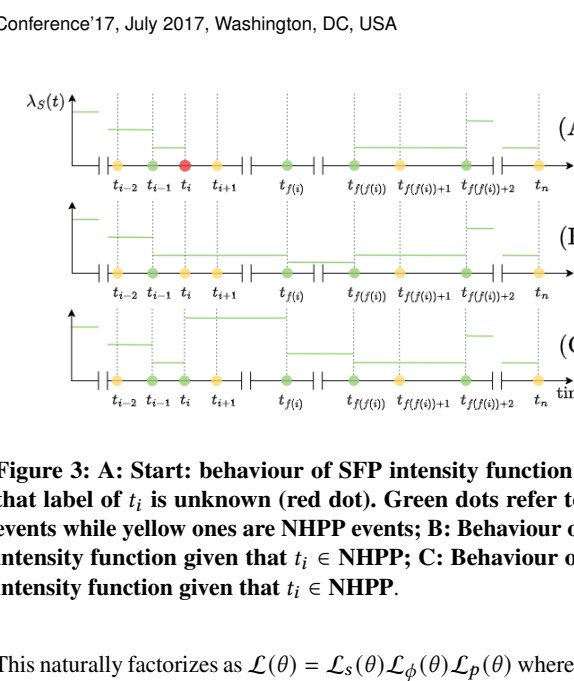

**Figure 3: A: Start: behaviour of SFP intensity function given that label of $t_i$ is unknown (red dot). Green dots refer to SFP events while yellow ones are NHPP events; B: Behaviour of SFP intensity function given that $t_i \in$ NHPP; C: Behaviour of SFP intensity function given that $t_i \in$ NHPP.**

This naturally factorizes as $\mathcal{L}(\theta) = \mathcal{L}_s(\theta)\mathcal{L}_\phi(\theta)\mathcal{L}_p(\theta)$ where $\mathcal{L}_s(\theta)$ $\mathcal{L}_\phi(\theta)$ and $\mathcal{L}_p(\theta)$ are, respectively, the components of the likelihood function (evaluated at $\theta$) corresponding to the SFP, MPP and NHP components: $\mathcal{L}_s(\theta) = \prod_{i=1}^n \lambda_s(t_i)^{z_i} e^{-\int_0^{t_n} \lambda_s(t)dt}$. $\mathcal{L}_\phi(\theta)$ and $\mathcal{L}_p(\theta)$ are defined similarly.

A key observation now is that the factors in (1) corresponding to the intervals $(0, t_{i-1}]$ and $(t_{f(f(i))}, t_n]$ do not depend on the values $\{z, m\}$, where $f(u) = \text{argmin}_j\{t_j | t_u < t_j \wedge z_j = 1\}$ denotes the index of the next SFP event after $t_u$. Indeed, whether $t_i$ is an SFP or Poisson event only influences the SFP intensity of the next two SFP events. Therefore, we can write equivalently:

$$\mathbb{P}(z_i = z, m_i = m | T, \theta_{-i}) \propto \mathcal{L}_s^i(\Omega_{z,m})\mathcal{L}_\phi^i(\Omega_{z,m})\mathcal{L}_p^i(\Omega_{z,m}), \quad (2)$$

where $\mathcal{L}_s^i(\Omega_{z,m})$ (resp. $\mathcal{L}_\phi^i(\Omega_{z,m})$ and $\mathcal{L}_p^i(\Omega_{z,m})$) corresponds to the component of the likelihood corresponding the SFP (resp. MPP, NHPP) and to the interval $[t_{i-1}, t_{f(f(i))})$ and $\Omega_{z,m} := \theta_{-i} \cup \{z, m\}$. Thus $\theta = \Omega_{z,m} \cup \Phi_i$ and we have

$$\mathbb{P}(z_i = z, m_i = m | T, \theta_{-i}) \propto \int_{\Phi_i \in F_{\Omega_{z,m}}} \mathcal{L}(\Omega_{z,m} \cup \Phi_i)d\Phi_i$$
$$\propto \mathcal{L}_s^i(\Omega_{z,m})\mathcal{L}_\phi^i(\Omega_{z,m}) \int_{\Phi_i \in F_{\Omega_{z,m}}} \mathcal{L}_p^i(\Omega_{z,m} \cup \Phi_i)d\Phi_i, \quad (3)$$

where we can write $\mathcal{L}_s^i(\Omega_{z,m})$ for $\mathcal{L}_s^i(\Omega_{z,m} \cup \Phi_i)$ for any $\Phi_i$ since $\mathcal{L}_s^i$ doesn't depend on $\Phi_i$ and $F_{\Omega_{z,m}}$ is the set of $\Phi_i$s compatible with the values of $T, M$ when $m_i$ is set to the index $m$: for instance, if $m_{i-1} = m_{i+1}$, then there are no transitions in the interval $[t_{i-1}, t_{i+1}]$, so $F_{\Omega_{z,m}} = \{\emptyset\}$. On the other hand, if $m_{i+1} - m_{i-1} = 1$ and $m = m_{i+1}$, $F_{\Omega_{z,m}}$ is the interval $[t_{i-1}, t_i)$.

To develop an intuition of how the label of $t_i$ affects $\mathcal{L}_s^i(\Omega_{z,m})$, we will demonstrate how to compute the intensities in the interval of interest. A similar explanation can be reproduced for $\mathcal{L}_\phi^i(\Omega_{z,m})$ and $\mathcal{L}_p^i(\Omega_{z,m})$. Consider Figure 3. By definition, given the parameter $\mu$, the SFP intensity depends solely on the two last SFP events. Therefore the computation of $\lambda_s(t)$ before $t_i$ only depends on the labels of the events before $t_i$ and, therefore, as they are known, such labels are not influenced by whether $t_i$ is a Poisson or a SFP event (note the green lines before $t_i$, Figure 3-A). Similarly, after $t_{f(f(i))}$ all the labels of the events are known and $\lambda_s(t)$ can be directly

computed (note the green lines after $t_{f(f(i))}$, Figure 3-A). However, the label of event $t_i$ impacts the value of $\lambda_s(t)$ between $t_i$ and $t_{f(f(i))}$. In Figure 3-B, we consider the case where $t_i \in$ Poisson. In this case, the only SFP shift will happen at $t_{f(i)}$, which is (by definition) the first SFP event after $t_i$. Observe that the intensity between $t_i$ and $t_{f(i)}$ remains the same as before $t_i$, with the next change occurring at $t_{f(i)}$. On the other hand, if $t_i \in$ SFP (see Figure 3-C), two shifts happen in the interval. The first one immediately after $t_i$ and the second one after $t_{f(i)}$.

Therefore, $\mathcal{L}_s^i(\Omega_{z,m})$ and $\mathcal{L}_\phi^i(\Omega_{z,m})$ can be computed directly. The integral $\mathcal{L}_p^i(\Omega_{z,m}) := \int_{\Phi_i \in F_{\Omega_{z,m}}} \mathcal{L}_p^i(\Omega_{z,m} \cup \Phi_i)d\Phi_i$ can be expressed as

$$C \prod_{j:t_j \in [t_{i-1}, t_{f(f(i))})} \left(\lambda_p(t_j | \Omega_{z,m})\right)^{1-z_j}$$
$$\times \mathcal{I}(\Omega_{z,m}, t_{i-1}, t_i, m_{i-1}, m) \times \mathcal{I}(\Omega_{z,m}, t_i, t_{i+1}, m, m_{i+1}). \quad (4)$$

The constant $C$ is independent of $m, z$, defined as

$$C = e^{t_{i-1}\lambda_p(t_{-i})} e^{-t_{i+1}\lambda_p(t_{+i})} e^{-\int_{t_{i+1}}^{t_{f(f(i))}} \lambda_p(t)dt},$$

and

$$\mathcal{I}(\Omega, t_b, t_e, m_b, m_e) = \int_{\mathcal{T}} e^{\sum_{j=m_b}^{m_e} \left(\lambda_{(j+1)} - \lambda_{(j)}\right)x_{(j-m_b+1)}}dx,$$

where $\mathcal{T} = \{x = (x_1, \ldots, x_{m_e-m_b}) | t_b \le x_1 \le x_2 \le \cdots \le x_{m_e-m_b} \le t_e\}$. The integrand in the definition of $\mathcal{I}$ is proportional to the likelihood of observing no Poisson event between $t_e$ and $t_b$ assuming $x_i$ is the $m_b + i$th transition for all $1 \le i \le m_e - m_b$. The integrals in the above equations can be computed using the strategy described in the Appendix of this paper. This concludes the explanation of the computation of $\mathbb{P}(z_i = z, m_i = m | T, \theta_{-i})$. Since $z_i, m_i$ are discrete random variables, it is then straightforward to sample from the corresponding distribution.

To complete our description of the Gibbs update which yields $\theta_i$, we must describe how to draw $\Phi_i$ from its conditional distribution assuming $\Omega_{z,m}$ is given. Note that given $\Omega_{z,m}$, $\Phi_i^-$ and $\Phi_i^+$ are independent, where $\Phi_i^- := \{\varphi_j | t_{i-1} < \varphi_j < t_i\}$ and $\Phi_i^+ := \{\varphi_j | t_i < \varphi_j < t_{i+1}\}$. The probability density function of $\Phi_i^-$ (resp. $\Phi_i^+$) is proportional to $\mathcal{I}(\Omega_{z,m}, t_{i-1}, t_i, m_{i-1}, m)$ (resp. $\mathcal{I}(\Omega_{z,m}, t_i, t_{i+1}, m_i, m_{i+1})$). To generate a sample from the distribution of $\Phi_i^-$ in practice ($\Phi_i^+$ is completely analogous), we make the following observations. For any interval $[a, b]$, let $N_{a,b} = \#(j : \phi_j \in [a, b])$ and $f_1 = (t_{i-1} + t_i)/2$. We have that the joint distribution of $(N_{t_{i-1}, f_1}, N_{f_1, t_i})$ evaluated at $(n_1, n_2)$ (with $n_1 + n_2 = m_{i+1} - m_{i-1}$) is proportional to $\mathcal{I}(\Omega_{z,m}, t_{i-1}, f_1, m_{i-1}, m_{i-1}+n_1)\mathcal{I}(\Omega_{z,m}, f_1, t_i, m_{i-1}+n_1, m_i)$. Thus, a sample can be drawn from it. We can continue to split the interval $[t_{i-1}, t_i)$ iteratively, choosing at each step how many $\varphi_j$s are on each side of each subinterval by drawing from the relevant discrete distributions. This can be done until only one $\varphi_i$ is in each interval, and its precise position can then be determined by a draw from its now one-dimensional probability distribution. This concludes the generation procedure for the $E$ step.

**M-Step:** Now, we will elucidate the process of maximizing the log-likelihood, which corresponds to the current estimate of the conditional distribution of $\theta$, over the parameter set $\{\mu, \Lambda\}$. The procedure described in the **E-Step** section allows us to draw $N_\theta$ samples $\{\theta_1, \theta_2, \cdots, \theta_{N_\theta}\}$ from the conditional distribution of $\theta$

given the current estimate of $\{\mu, \Lambda\}$. We then update $\mu$ via the formula $\hat{\mu} = \left( \sum_{j=1}^{N_\theta} \text{argmin}_\mu \log(\mathcal{L}_s(\theta_j)) \right) / N_\theta$, where the likelihood minimization steps are performed via binary search. Note that $\mu$ is an easy parameter to estimate as it affects the whole interval, thus $\text{argmin}_\mu \mathcal{L}_s(\theta_j)$ is already a good estimate even for a single value of $j$.

Regarding the set of parameters $\Lambda$, a key observation is that $\mathcal{L}_s(\theta)$ and $\mathcal{L}_\phi(\theta)$ are independent of $\Lambda$, which allows us to perform the maximization over $\mathcal{L}_s(\theta)$ alone. Let $U_j(\theta) = \sum_j 1_{t_j \in [\bar{\varphi}_j, \bar{\varphi}_{j+1}] \wedge z_j = 0}$. The part of $\log(\mathcal{L}_s(\theta))$ which depends on $\lambda_j$ is $-(\bar{\varphi}_{j+1} - \bar{\varphi}_j) + U_j(\theta) \log(\lambda_j) - \log(U_j(\theta))$. Averaging over all values of $\theta$ and optimizing over $\lambda_j$, we immediately obtain the following formula for the $\lambda$s:

$$\hat{\lambda}_i = \frac{\sum_{j=1}^{N_\theta} U_i(\theta_j)}{\sum_{j=1}^{N_\theta} (\bar{\varphi}(\theta_j)_{i+1} - \bar{\varphi}(\theta_j)_i)}, \tag{5}$$

i.e. we are treating the observations of the Poisson events on the intervals corresponding to $\lambda_j$ as if they came from a fixed homogeneous Poisson process. This is valid since the value of $\Lambda$ only influences the Poisson likelihood component. Algorithmic details and a complexity analysis of our method can be found in the Appendix.

## 4 EXPERIMENTS

In this section, we present our experiments conducted with real-world RSEs collected from various web systems. Additionally, we conduct synthetic data experiments (please refer to the Appendix) to validate our model under different ground truth scenarios and evaluate the effectiveness of our EM algorithm's derivation in recovering the model's underlying parameters.

**Datasets:** We showcase **BPoP**'s utility on 11 real-world datasets involving RSEs from diverse web systems across various domains, detailed in Table 1. In **AskMe**, **MetaFilter**, and **MetaTalk**, the time-series represent online discussion forum topics, with events being timestamps of comments[3]. In **Digg**, each time-series corresponds to a news post, and events are the 'diggs,' similar to Facebook 'likes'[4]. In **Enron**, events are timestamped emails associated with e-mail accounts[5]. The **Github** dataset is split into two parts: **Github (Users)** and **Github (Projects)**. The former records user activities across different projects, while the latter documents user activities on specific projects. **Google Trends**, time series correspond to the fraction of YouTube views over time (only USA users). Each topic is related to famous people such as singers and politicians, which were defined and collected by the authors. In **Twitter**, event timestamps correspond to tweets with specific hashtags. For **Youtube**, each time series represents a YouTube video, with events as timestamps for user comments. Finally, the **Yelp** dataset contains timestamps of user ratings for various restaurants.

Table 1 shows the total number of RSEs, the average number of events, as well as the average of the indices *absolute stability* $\kappa$ and *relative stability* $\tilde{\kappa}$ (defined further in this section) for each dataset considering the entire observed time interval of each time series. Out of the total population, 91% of individuals exhibited a value of $0.05 < P_{NHPP} < 0.95$: this suggests a combination of stable behavior and a curious audience. Among those, 21% have

---

[3] Available at http://stuff.metafilter.com/infodump
[4] Available at: http://digg.com
[5] Available at: https://www.cs.cmu.edu/~./enron/

at least one transition, i.e., their *stable audience* changed during the analyzed period, an assumption that motivated the conception of **BPoP**. The fifth column ($|\Phi|$) in Table 1 shows the the average number of transitions for each dataset among the individuals that had transitions. In total, we analyzed more than 78 million events.

**Disentangling Stable and Ephemeral Audiences:** We demonstrate the effectiveness of our disentangling method using data from the `#ACL` Twitter hashtag during the *Austin City Limits* (ACL) festival in 2009. ACL is an annual three-day music and art festival held in Austin, Texas, USA, attracting over 130 bands, with around 65,000 daily attendees. In 2009, ACL promoted the "The Sound and the Jury competition (SJC)," a virtual band contest offering a festival slot to the winner. Our model, relying solely on event timestamps related to the Twitter hashtag #ACL (Figure 5, top left), effectively separates the series of events into *stable audience* (NHPP) and *curious* (SFP) components. Notably, in this context, the audience is not tied to the festival itself but rather to the related **hashtag** in the period *before, during*, **and** *after the festival*. The *stable audience* (of the hashtag) comprises dedicated music fans who regularly engage with the Twitter feed, while the *curious audience* consists of individuals intrigued by the festival. The first component (Figure 5, bottom left) maintains a constant event arrival rate between transitions $\varphi_1, \varphi_2, \ldots$ (indicated by vertical lines). This rate $\lambda_p(t)$ signifies the *stable audience* rate at a given time $t$. In our model, bursts (Figure 5, top right) are associated with the intensity of the SFP, $\lambda_s(t)$ (Figure 5, bottom right), representing the *curious* audience. Importantly, we acknowledge that the rate $\lambda_p(t)$ is not stationary, as bursts (modeled by the SFP component) are triggered by external or internal events related to the topic. These factors not only generate short bursts of intense activity but also lead to enduring changes in the topic's discussion dynamics. Examples of such incidents could include retweets by prominent celebrities or the passing of influential figures related to the topic, both of which can alter the composition and behavior of the *stable audience*.

These facts can be observed in the behavior of the hashtag #ACL and its disentangled representation provided by **BPoP**. In the pre-SJC period, the arrival rate of the events was very low: at this time, only hard-core fans were actively posting tweets, namely the *stable audience*. This period of calm was disrupted by the SJC campaign period. The SJC campaign (highlighted in purple) was a short period of time during which the bands involved in the contest together with their fans (the *curious*) posted a large number of tweets asking users to vote for them. In addition to the short burst of tweets asking for votes, this effect altered the topic dynamic: now, it was not only the hard-core fans of the festival, but also the bands' supporters which were active in the social network. This effect was to continue until the end of the first round, and the announcement of the TOP-20 bands, which would be selected to compete in the next stage. After this announcement, since the number of bands in the contest has decreased, we expect a decrease in the number of supporters, resulting in fewer hashtag users and tweets. From **BPoP**, we indeed observe a transition at this event and a decreased *stable audience* (NHPP) rate afterwards. The new rate remained constant until the ACL event itself. Finally, a huge burst of events occurred during the festival, which was correctly modeled by our model as mostly part of the SFP process, which represents the *curious*. Our model also detects a transition at this event, and the rate goes back to pre-SJC

**Table 1: Fitting and characterization of the datasets**

| | #RSE | n | $\kappa$ | $\tilde{\kappa}$ | $|\Phi|$ | $R^2_{\text{BPoP}}$ | $R^2_{\text{BP}}$ | $R^2_{\text{Hawkes}}$ |
|---|---|---|---|---|---|---|---|---|
| **AskMe** | 490 | 133 | 0.36 | 0.45 | 1.03 | **0.8514 ± 0.11** | 0.6204 ± 0.10 | 0.3211 ± 0.15 |
| **Digg** | 974 | 122 | 0.31 | 0.45 | 1.74 | **0.8944 ± 0.10** | 0.7231 ± 0.09 | 0.4824 ± 0.17 |
| **Enron** | 147 | 1589 | 0.59 | 0.49 | 2.21 | **0.9449 ± 0.07** | 0.9178 ± 0.06 | 0.5954 ± 0.25 |
| **GitHub(U)** | 40385 | 675 | 0.76 | 0.50 | 1.40 | **0.9565 ± 0.03** | 0.9526 ± 0.04 | 0.8876 ± 0.10 |
| **GitHub(P)** | 35085 | 696 | 0.77 | 0.50 | 1.39 | **0.9570 ± 0.03** | 0.9523 ± 0.04 | 0.8853 ± 0.10 |
| **G. Trends** | 579 | 2975 | 0.66 | 0.48 | 2.09 | **0.9632 ± 0.07** | 0.9596 ± 0.01 | 0.7973 ± 0.29 |
| **MetaFilter** | 8249 | 172 | 0.42 | 0.43 | 1.23 | **0.8931 ± 0.09** | 0.7123 ± 0.11 | 0.3906 ± 0.18 |
| **MetaTalk** | 2465 | 203 | 0.43 | 0.49 | 1.26 | **0.9176 ± 0.08** | 0.7921 ± 0.11 | 0.4535 ± 0.20 |
| **Twitter** | 18888 | 1142 | 0.71 | 0.50 | 1.60 | **0.9469 ± 0.05** | 0.8882 ± 0.12 | 0.8098 ± 0.23 |
| **Yelp** | 1931 | 128 | 0.22 | 0.38 | 1.34 | 0.9193 ± 0.13 | **0.9411 ± 0.08** | 0.8402 ± 0.14 |
| **YouTube** | 250 | 3241 | 0.59 | 0.49 | 1.90 | **0.9720 ± 0.02** | 0.9696 ± 0.01 | 0.7010 ± 0.18 |

levels afterwards. After the festival ends, the audience is once again composed of the *stable audience* only.

Our model core hypothesis is that short bursts of activity are likely to simultaneously change the *stable audience* constitution because they share a common cause: topic related unusual incidents. We model this by making the transition occurrence rate equal to $c\lambda_s(t)$ or proportional to the arrival rate of atypical (SFP) events ($\lambda_s(t)$), where $c$ is small. SFP events appear also during non-bursty periods. **BPoP** allows for the possibility of transitions occurring also during calm periods, not only during bursts, and our algorithm is able to detect such transitions.

**Absolute and Relative Stability:** After learning the component intensities $\lambda_s(t)$ and $\lambda_p(t)$ we can contrast their absolute and relative influence on the observed events. We define two indexes, both in the interval $[0, 1]$:

$$\kappa = \int_a^b \frac{\lambda_p(t)(b-a)^{-1}}{\lambda_p(t) + \lambda_s(t)} dt \text{ and } \widetilde{\kappa} = \frac{\int_a^b \lambda_p(t) dt}{\int_a^b (\lambda_p(t) + \lambda_s(t)) dt}. \quad (6)$$

The *absolute stability* $\kappa$ tells us the importance of the *stable audience* averaged over the time interval $[a, b]$, whilst the *relative stability* $\widetilde{\kappa}$ describes the proportion of the activity in the interval $[a, b]$ that is assigned to the *stable audience*. The plot on the left hand side of Figure 4 shows the average $\tilde{\kappa}$ versus the average $\kappa$ for each of the real datasets. Consider the two pairs associated with GitHub: $\kappa \approx 0.8$ but $\tilde{\kappa} \approx 0.5$. This shows that while the *stable audience* (NHPP component) dominates most of the time, only half of the activities are carried out by them, i.e., the other half comes from the *curious*.

The plots in the right hand side of Figure 4 show the parameters $\lambda_p$, $\kappa$ and $\widetilde{\kappa}$ calculated in each year separately. The time series are USA Youtube searches for five artists out of the Google Trends dataset, each artist shown in a column of plots. The data was collected between Jan $1^{st}$ 2011 to Dec $31^{st}$ 2020.

The artists were selected to show widely different composition of the latent processes or, in other words, how much of their audiences are composed by the *curious* and by the *stable audience*. Alex Claire is an English singer and his biggest hit was released in 2011. Despite the huge success at this time, it was not enough to keep a stable American audience on Youtube. This fact is confirmed by the observation that the coefficients $\lambda_p$, $\kappa$, and $\widetilde{\kappa}$ have low values for the

entire period, characterizing a small *stable audience*. It is almost a pure SFP process, or an audience composed mostly by the *curious*.

The time series related to the artists Redfoo, Bridgit Mendler and Billie Eilish exhibit a mixture between bursty and calm periods. Redfoo is able to maintain a significant *stable audience* during the whole time period, while Mendler and Ellish see their *stable audience* decrease and increase over time, respectively. In absolute numbers, Bridgit Mendler has the largest *stable audience*, while Redfoo has the smallest. Interestingly, despite the evident decrease in Mendler's *stable audience*, she remains quite popular when compared to her counterparts. The *absolute stability* $\kappa$ shows that the presence or absence of bursts modifies the relative importance of the NHPP in the composition of the model, which suggests trends and seasonality.

Lastly, Stacey Q is an American pop singer who was popular in the 80's. As we can observe from the constant $\lambda_p$, she still enjoyed a stationary amount of attention from a *stable audience* between 2011 and 2020. Her time series did not exhibit any burst during the whole period which explains the high $\kappa$. Her audience is probably composed of long-time fans who do not have the necessary engagement to produce such changes. It is an almost pure Poisson process.

## 5  GOODNESS OF FIT

**Baselines: BPoP** is a model that relies only on the observed event timestamps. We compare our model with two other similar models.

*Hawkes processes (HP)* [20, 30, 40–42] are a class of self-exciting processes which are widely used for modeling web communications. The Hawkes process model assumes that any event increases the probability of additional events. Its conditional intensity is $\lambda(t|\mathcal{H}_t) = \lambda + \sum_{t_i < t} K(t - t_i)$, where $K(x) > 0$ is the kernel function, which satisfies $\int_0^\infty K(x)dx < 1$, to ensure stationarity.

*BuSca* [18]: similarly to our model **BPoP**, BuSca is a mixture process involving a Poisson process and a self-exciting process. The conditional intensity of the BuSca model is given by $\lambda(t|\mathcal{H}_t) = \lambda + \frac{1}{\Delta_t + \mu/e}$, where $\lambda \geq 0$ and $\mu > 0$ are constants and $\Delta_t$ is the last SFP interval before $t$. However, **BPoP** and BuSca have two important differences. Firstly, BuSca does not assume any changes in the $\lambda$ Poisson rate, which corresponds to the unrealistic assumption that the Poisson component of the behavior is constant and occurs indefinitely. Secondly, there is no interaction between the two components: the behavior of the Poisson component is not influenced

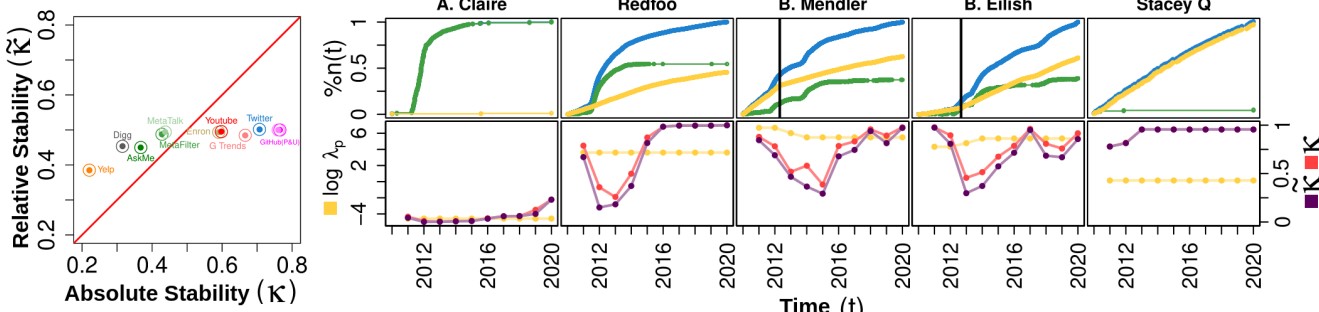

**Figure 4: Left: Scatterplot of the average $\tilde{\kappa}$ versus the average $\kappa$ for each of the real datasets. Right: Each column of the plots corresponds to the Google Trend time series associated with one of 5 artists. The top row shows the raw counts. The bottom plots show the yearly average of $\log(\lambda_p(t))$ (yellow line), $\kappa$ (magenta line), and $\tilde{\kappa}$ (red line).**

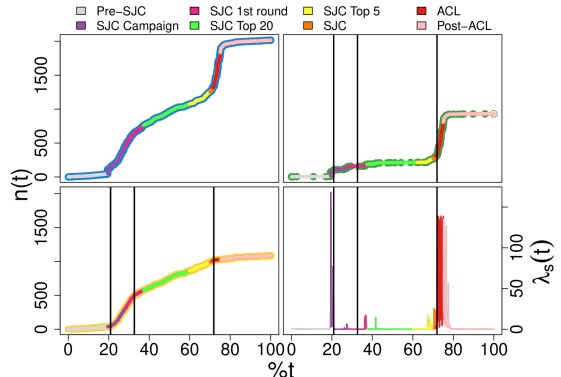

**Figure 5: Analysis of the `#ACL` Twitter hashtag associated with the 2009 *Austin City Limits* (ACL) festival. Top left: cumulative events; Top right: *curious* (burst) component; Bottom left: *stable audience* (NHPP) component; Bottom right: intensity of the SFP.**

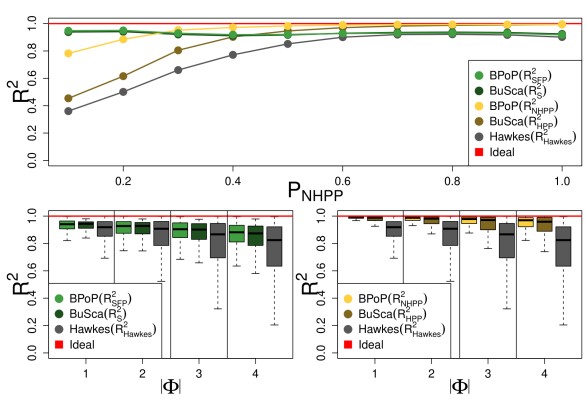

**Figure 6: Top: fitting performance (determination coefficient) as a function of the proportion $|PP|/n$ of Poisson events. Bottom: fitting performance as a function of the number of transitions.**

by the self-exciting component and vice-versa. In contrast, **BPoP** allows for significant changes in the Poisson rate. More importantly, these changes are motivated by the SFP events: they influence the presence of transitions between different Poisson regimes. This is a defining characteristic of our model, a more realistic assumption, and a significant source of added mathematical and computational difficulty.

**Metrics:** To assess the Hawkes Process method's performance we used the random time change theorem to transform a HP into a unit rate Poisson process (see [31]). After the transformation we computed the determination coefficient $R^2_{Hawkes}$ corresponding to the linear regression problem predicting the cumulative number of events $N(t)$ for all $t$s in the transformed process. Similarly, for BuSca, we computed $R^2_S$ (SFP) and $R^2_{HPP}$ (homogeneous Poisson process) for the disentangled processes (see [18]) and computed the final coefficient $R^2_{BP} = (R^2_S + R^2_{HPP})/2$.

To check the goodness of fit of **BPoP**, we first output the $\{\hat{Z}, \hat{\Phi}\} \subset \theta_i$, where $i = \text{argmax}_j\{\mathcal{L}(\theta_j) ; 1 \le j \le N_\theta\}$, which allows us to disentangle the NHPP from the SFP. For the SFP fitting, we took the inter-event times sample and built the empirical cumulative

distribution function $\mathbb{F}(t)$ leading to the odds-ratio function $OR(t) = \mathbb{F}(t)/(1-\mathbb{F}(t))$. Then, we computed the $R^2_{SFP}$ coefficient of the linear regression problem predicting the cumulative number of events $N(t)$ versus the $OR(t)$ (see [33]).

The computation of $R^2_{NHPP}$ requires some explanation. Let $\bar{\Phi}$ be the list of the elements of $\{0 \cup \Phi \cup t_n\}$ ordered from smallest to largest, so that $\bar{\varphi}_0 = 0$, $\bar{\varphi}_{m_t+1} = t_n$. For all $i \in \{0, 1, \cdots, |\bar{\Phi}| - 1\}$, we construct the set $T_i = \{t_j | \bar{\varphi}_i < t_j < \bar{\varphi}_{i+1}\}$ and then estimate $R^2_{NHPP_i}$ as the determination coefficient corresponding to the linear regression problem predicting $N(t)$ from $t$ on the interval $[\bar{\varphi}_i, \bar{\varphi}_{i+1})$ with the datapoints obtained from $T_i$. Finally, we compute $R^2_{NHPP}$ as the weighted average of the $R^2_{NHPP_i}$s. The weight is a multiple of the respective interval. More formally, $R^2_{NHPP} = \sum_i (\bar{\varphi}_{i+1} - \bar{\varphi}_i)R^2_{NHPP_i}/t_n$. Finally, we compute $R^2_{\mathbf{BPoP}} = (R^2_{SFP} + R^2_{NHPP})/2$.

All $R^2$ coefficients vary between 0 (*worst case*) and 1 (*best case*). Table 1 shows the goodness-of-fit statistics (average and standard deviation) for **BPoP** and for the baselines, grouped by dataset. **BPoP** surpasses the Hawkes process method in all datasets considered. It also consistently outperforms BuSca (better fitting in 10 out of 11 datasets). Indeed, the high concentration of the $R^2_{\mathbf{BPoP}}$ (as $R^2_{SFP}$ and $R^2_{NHPP}$, in Figure 6) statistics close to the maximum

value of 1 shows that our model can accurately fit the time series considered, as well as disentangle the mixed process into its two hidden components (NHPP and SFP).

**Results:** Figure 6 shows how our disentangled models behave under different regimes. To construct the top graph, we computed $P_{NHPP}$ and rounded the value considering the range $\{0, 0.1, 0.2, \cdots, 1\}$. The extremes correspond, respectively, to a pure Poisson processes and a pure SFP. We can observe that our model improves significantly when the mixture is dominated by the bursty behavior. With respect to the number of transitions $|\Phi|$, the boxplots on the bottom show that **BPoP** outperforms the baselines across the whole spectrum, with the performance increasing with the number of transitions.

## 6 RELATED WORK

**Human Communication Dynamics:** Characterizing the dynamics of human communication on the web has significant implications for various applications, including trend detection, clustering, anomaly detection, and popularity prediction[34, 40, 41, 47]. This research is inspired by a substantial body of work focused on predicting the popularity of online content, such as YouTube videos, hashtags, and forum posts[9, 20, 24–26, 29]. The primary goal is to estimate the total number of events associated with a given item. At first, Crane and Sornette posited two primary mechanisms for the occurrence of viewing activity: random occurrences influenced by external factors such as featuring, or internally driven through sharing[19]. However, recent studies have identified additional factors influencing popularity, such as content quality[4, 5], item metadata[6], item age[7], recommendation algorithms, ranking in keyword-based queries[8], and social network effects[4, 10].

More recently, Yu et al. [24] have introduced a phase representation model for online videos, extending Crane and Sornette's endogenous growth and exogenous shock model. They discovered that online videos undergo multiple stages of popularity fluctuations over several months. This finding received further support from Rizoiu et al. [9], who identified a strong correlation between external promotion and online video popularity. Their research highlighted the substantial impact of external attention and promotion on video popularity. Additionally, Gleeson et al. [25] emphasized the significance of recent popularity over cumulative popularity in the adoption of Facebook apps.

**RSEs Modeling:** Human activity on the web displays a broad spectrum of unpredictability, ranging from complete randomness [19, 35–37] to high correlation and burstiness [40, 41, 43, 48, 49]. These diverse patterns have prompted the adoption of point process stochastic models, which provide statistical frameworks for comprehending sequences of random events [31, 32]. In principle, these models can be employed to estimate the audience size (*fanbase*) of online items. However, existing models are not well-suited for this specific task. Poisson processes (PPs) [36, 37] are suitable when events arrive regularly at a fixed rate, allowing for stable audience estimation. While a significant portion of online items can be accurately described by such a simple model [10, 11, 17, 19], PPs have limitations. For example, Malmgren et al. [37] introduced a non-homogeneous Poisson process model that accounts for circadian cycles with varying rate $\lambda(t)$, but it lacks the self-exciting property. This means that the probability of observing an event at a small time interval $[t, t + \Delta t)$ does not depend on previous events within that interval. This limitation hampers PPs from effectively capturing event bursts observed in real-world data.

On the other hand, while self-exciting processes effectively capture correlations between consecutive events responsible for activity bursts in real data, existing methods often overlook the time-varying nature of the fanbase. Hawkes processes [20, 30, 40–42], one of the most widely used models, maintain a constant baseline rate and incorporate event history via a conditional intensity formula: $\lambda(t|\mathcal{H}_t) = \lambda + \sum_{t_i < t} K(t - t_i)$, where $K(x) > 0$ is typically a decreasing exponential kernel. Hawkes processes promote bursty behavior and fall into the category of pure self-exciting models with a constant background intensity. They offer an alternative to pure Poisson processes, with nuanced mathematical properties and applications. However, they cannot capture changes in background intensity, a feature addressed in our work.

The recent literature reflects a growing interest in exploring alternatives to the widely-used Hawkes-based processes for modeling self-exciting point process data. For instance, Etesami et al. [50] and Trouleau et al. [51] have applied variational inference algorithms to fit Bayesian models for multivariate self-feeding processes, enabling the analysis of real-world communication dynamics. Moreover, Noorbakhsh and Rodriguez [52] have introduced a novel class of Gumbel-max point processes specifically designed to address causal issues in point process modeling.

Our work consistently advocates the adoption of self-feeding processes as a compelling alternative to widely-used Hawkes-based models. Previous research, including [18, 43, 53, 54], strongly supports this approach. Its appeal lies in its simplicity and its ability to accurately capture the short-memory and power-law behavior common in real-world data. Self-feeding processes produce point patterns characterized by bursts of intense activity followed by periods of low activity, aligning well with real-world observations. By applying our model to another real case, we aim to demonstrate its effectiveness as a competitive alternative for modeling self-exciting point processes. Finally, Alves et al. [18] employed a Wold process to model social media events effectively. However, this model, with a constant background rate, presents significant training challenges due to multiple approximations required for EM algorithm expectations. Our novel model addresses these limitations by accurately mimicking event bursts while efficiently capturing time-varying background rates, providing a more realistic representation of our fanbase dynamics.

## 7 CONCLUSION

In this article, we presented **Burst-induced Poisson Process** (**BPoP**), a model that separates stable and curious media audiences. **BPoP** combines an SFP for viral thread bursts (representing the *curious* audience) with a non-homogeneous Poisson process for regular user behavior (the *stable audience*). These components interact, and we develop a tailored EM algorithm to address this complexity. Our model excels in identifying audience dynamics in both synthetic and real data.

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

# APPENDIX

## A SYNTHETIC EXPERIMENTS

**BPoP** is a *generative model* that combines an SFP to represent the *curious* audience and an NHPP to represent the *stable audience*, both interacting with each other. While the complete generation procedure is originally detailed in the main paper's model description, we're providing a concise summary in this appendix for ease of reference.

**Generation procedure:** consider that, at time $t$, the history of the process is composed of the observed event timestamps $\{t_1, t_2, \ldots\} < t$, unobserved labels $\{z_1, z_2, \ldots\}$ as well as unobserved MPP events $\{\varphi_1, \varphi_2, \ldots\}$, which represent the *transitions*. We use the convention that $z_i = 0$ if $t_i \in$ NHPP and $z_i = 1$ if $t_i \in$ SFP. Define the following three intensity functions: (1) the SFP intensity $\lambda_s(t) = 1/[(t_{g(t)} - t_{g(g(t))}) + \mu/e]$ where $g(u) = [\max(t_i : z_i = 1 \quad \wedge \quad t_i < u)]_+$ denotes the last SFP event before $t$, with the convention that $g(t) = 0$ if $\nexists i : t_i \leq t \wedge z_i = 1$ and $\mu > 0$ is the SFP parameter; (2) $\lambda_\Phi(t) = c\lambda_s(t)$, where $c \in [0, 1]$ is a parameter that controls the NHPP transition sensitivity; and (3) $\lambda_p(t) = \lambda_{m_t}$, where $\Lambda = \{\lambda_0, \lambda_1, \lambda_2, \cdots\}$ is an infinite set of positive numbers (parameters) and, $m_t = \sum_{j=1}^{n_\phi} 1_{\varphi_j < t}$. Similarly we can define $\lambda_s^+(t) = \lim_{\delta \to 0} \lambda_s(t + \delta)$ (resp. $\lambda_\phi^+(t)$ and $\lambda_p^+(t)$) as the intensity of the SFP (resp. MPP and NHPP) immediately after $t$. Thus, to generate the next time stamp, we first generate three exponential variables $E_s$, $E_\phi$ and $E_p$ with intensities $\lambda_s^+(t)$, $\lambda_\phi^+(t)$ and $\lambda_p^+(t)$ respectively. Then, the next event will take place at $t + E$, where E$=\min(E_s, E_\phi, E_p)$, and it will belong to the SFP (resp. NPHH, MPP) component if $E = E_s$ (resp. $E_\phi$, $E_p$). Likewise, we continue generating the rest of the process from time $t + E$.

**Experimental setup:** Let $P_{NHPP} = \sum_{i=1}^{n} (1 - z_i)/n$ be the proportion of observed events that belong to NHPP. We chose sets of $\mu, \Lambda$ and $c$ corresponding to estimated values of $(\mathbb{E}(n), \mathbb{E}(|\Phi|), \mathbb{E}(P_{NHPP}))$ in the set $\{500, 750, 1000\} \times \{0, 1, 2\} \times \{0, 0.25, 0.5, 0.75, 1\}$. In the case where transitions are present, we considered only the cases in which the expected number of events of both processes (SFP and NHPP) is greater than 0. For example, the tuple $(500, 1, 0)$ was not considered since we would expect only one transition and zero NHPP events. Concerning the parameter $\Lambda$ we selected values such that $\forall i$ $\min(\lambda_i, \lambda_{i+1})/\max(\lambda_i, \lambda_{i+1}) = 1/3$.

For each tuple $(\mathbb{E}(n), \mathbb{E}(|\Phi|), \mathbb{E}(P_{NHPP}))$, we conducted 50 simulations (we use the generation procedure explained above) and assessed our methods via two metrics. To assess our method's ability to recover the ground truth model accurately given the observations, we aggregate the relative difference between the total intensities corresponding to our recovered labels and the ground truth labels:

$$\delta(\theta, \hat{\theta}) = \int_0^{t_n} \frac{|(\lambda_s(t|\theta) + \lambda_p(t|\theta)) - (\lambda_s(t|\hat{\theta}) + \lambda_p(t|\hat{\theta}))|}{\lambda_s(t|\theta) + \lambda_p(t|\theta)} dt. \quad (7)$$

The reason we do this instead of simply counting the proportion of correct labels is as follows. Correctly classifying the timestamps is both more difficult and less interesting inside a burst compared to calm periods. Furthermore, the parametrization of the model could present some redundancy, in which case very different parameter combinations could correspond to similar point processes. On the other hand, $\delta(\theta, \hat{\theta})$ is far less sensitive to the label assignments in a short bursty period, but a small value of $\delta(\theta, \hat{\theta})$ still indicates excellent performance. Indeed, it shows that the model accurately represents the position of the set of observations in the probability

space and would perform well at predicting the positions of further observations if they had been left unobserved.

The second metric simply aims at verifying convergence. We evaluate the log-likelihood $\log(\mathcal{L}(\hat{\theta}))$ at our model parameters (and at one high-likelihood draw of the conditional labels), and compare it with the log-likelihood evaluated with the ground truth parameters and labels $\log(\mathcal{L}(\theta))$.

We report the results of our experiments evaluated with both metrics in Figure 7. The box plot shows that our method has a strong ability to recover the underlying components (SFP and NHPP) based only on the observed timestamps. Larger values of the number of events ($n$) correspond to smaller values of $\delta(\theta, \hat{\theta})$. Mixtures with higher $P_{NHPP}$ tend to produce fewer bursts and therefore have a more uniform behavior over the whole observed period. Consistently with this, we observe that larger values of $P_{NHPP}$ correspond to smaller values of $\delta(\theta, \hat{\theta})$. The number of transitions ($|\Phi|$) has a lower impact in comparison to the other parameters', though smaller values of $\delta(\theta, \hat{\theta})$ are associated with fewer transitions. The rightmost graph in Figure 7 shows that $\log(\mathcal{L}(\hat{\theta}))$ is systematically close to $\log(\mathcal{L}(\theta))$, and even surpasses it in more than 80% of the cases. Both metrics' behavior jointly indicate that our method can accurately recover the ground truth based only on the timestamps of the observed mixture process.

# B ALGORITHMIC DETAILS AND COMPLEXITY ANALYSIS

---
**Algorithm 1 BPoP**

**INPUT** $T$, $k^*$, $N_\theta$ and $N_{Gibbs}$
**OUTPUT**: $\mu$ and $\Lambda$

---
1: $n \leftarrow |T|$
2: $\mu, \Lambda, \theta_1, \theta_2, \cdots, \theta_{N_\theta} \leftarrow \text{warmStarts}(T, k^*, N_\theta)$
3: **while** Not converged **do**
4:     **for** $i \in \{1, 2, \cdots, N_\theta\}$ **do**
5:         $\theta_i = \text{GibbsSampler}(\theta_i, \mu, \Lambda, N_{Gibbs})$
6:     **end for**
7:     $\mu, \Lambda = \text{M-STEP}(\theta_1, \theta_2, \cdots, \theta_{N_\theta})$
8: **end while**
9: **return** $\mu, \Lambda$

---

Algorithm 1 describes how to compute $\mu$ and $\Lambda$ for a fixed parameter set $k^*$, $N_\theta$ and $N_{Gibbs}$. The last parameter controls how many updates on the latent variables components need to be performed during the E-STEP. In practice, we can decrease $N_{Gibbs}$ gradually during the EM-Algorithm execution to speed up convergence. Note that we set $N_{Gibbs} = O(n)$ so that each time stamp is updates a constant number of times.

$N_\theta$ can be adapted depending on the available computer resources. However a small number proved to be sufficient. We choose a small $k^*$ larger than the maximum number of transitions we expect and for each $k \in \{0, 1, \cdots, k^*\}$ we draw $N_\theta/(k^* + 1)$ samples of $\theta$. In the E-STEP, to prevent the model from getting stuck in low-likelihood regions, we performed likelihood-based re-sampling: after a several iterations, we replace the current estimate of the set $\{\theta_1, \ldots, \theta_{N_\theta}\}$ by a set of $N_\theta$ elements drawn with replacement from that set with probabilities proportional to the likelihoods.

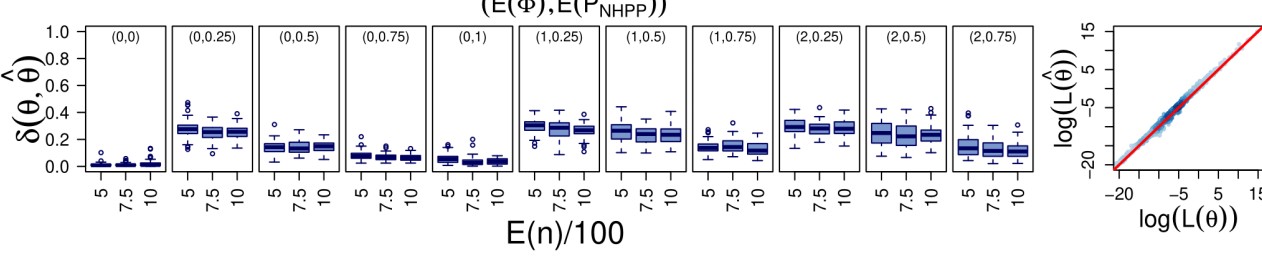

$(\mathbb{E}(\Phi), \mathbb{E}(P_{NHPP}))$

**Figure 7: Summary of the results of our experiments with synthetic data. Left: $\delta(\theta, \hat{\theta})$ distribution grouped by $(\mathbb{E}(|\Phi|), \mathbb{E}(P_{NHPP}))$. The $x$-axis shows the expected number of observed events. Right: plot of $\log(\mathcal{L}(\theta)) \times \log(\mathcal{L}(\hat{\theta}))$ with the $y = x$ line in red.**

**Warm-starts for $\theta$ and parameter initialization:** we set the initial number of transitions $k = |\Phi|$ as a fixed hyperparameter. Then, we first divide the interval $[0, t_n)$ into $n_k$ sub-intervals $I_1, \ldots, I_{n_k}$ of size $t_n/n_k$ (here $k < n_k \ll n$), i.e. $I_i = [(i-1)t_n/n_k, it_n/n_k)$. For each $i \le n_k$, we define $u_i = \sum_{j \le n} 1_{t_j \in I_i}$, the number of events belonging to interval $I_i$. Then we sample $k$ sub-intervals $I^1, I^2, \ldots, I^k$ without replacement with $\mathbb{P}(I_i) \propto u_i$. We then sample one timestamp from each interval $I^1, I^2, \ldots, I^k$, uniformly, and use the resulting set $\Phi$ of $k$ timestapms as the set of our transitions.

To estimate $\Lambda$ we divide the interval $[0, t_n)$ in $k + 1$ contiguous sub-intervals according to the transitions $\Phi$. Let $\bar{\Phi} = \{\bar{\varphi}_0, \bar{\varphi}_1 = \varphi_1, \ldots, \bar{\varphi}_m = \varphi_m, \bar{\varphi}_{m+1} = t_n\}$ be the list of the elements of $\{0 \cup \Phi \cup t_n\}$ ordered from smallest to largest starting with $\bar{\varphi}_0 = 0$, and for each $0 \le j \le k + 1$ we further divide the interval $[\bar{\varphi}_j, \bar{\varphi}_{j+1})$ into $n_k$ contiguous sub-intervals $I_0^j, I_1^j, \ldots, I_{n_k-1}^j$ of equal length. After that we compute, for each $j$ and $i$, the quantity $u_i^j = \sum_{o \le n} 1_{t_o \in I_i^j}$, and sample one sub-interval $I_v^j$ with $\mathbb{P}(v = i) \propto 1/u_i^j$ (except when $u_i^j = 0$ in which case $\mathbb{P}(v = i) = 0$ [6]) and set $\lambda_j = (u_{j_v} n_k)/(\bar{\varphi}_{j+1} - \bar{\varphi}_j)$.

To estimate $\mu$ and $Z$, we can now calculate $M := \{m_1, m_2, \cdots, m_n\}$, where $m_i = \sum_{j=1}^{n_\phi} 1_{\varphi_j < t_i}$. Let $p_i = \min(\lambda_{m_i}(\varphi_{m_i+1} - \varphi_{m_i})/|\Delta_i|, 1)$, where $\Delta_i = \{t_l | \bar{\varphi}_{m_i} < t_l < \bar{\varphi}_{m_i+1}\}$ and draw $z_i \sim$ Bernoulli$(1 - p_i)$. Thus, we can now estimate $\mu$ based on our estimated sample of the underlying SFP process $\{t_i | z_i = 1\}$, i.e. $\mu = $ median$(\{t_i | z_i = 1\})$We now have a value for the parameters $\Lambda, \mu$ and the latent variables $\theta = Z \cup \Phi \cup M$, which concludes the warm start and initialization phase.

**Complexity analysis:** The execution time of the algorithm is highly dependent on the observed data. Nevertheless, if we restrict the number of iterations of the EM algorithm to $N_{EM} \ll n$, our algorithm has complexity $O(N_{EM} N_{Gibbs} n)$ (indeed, each component of the likelihood calculations involved in the computation of the conditional probabilities consists of a sum over each event in the interval $[t_{i-1}, t_{f(f(i))})$). In the worst case, this interval is the whole of $T$ and the computation complexity for one Gibbs iteration is $O(n)$).

## C EXPLICIT COMPUTATION OF THE INTEGRAL $\mathcal{I}$

There exists an explicit formula for the multiple integral

---

[6]The idea is to avoid picking an interval with a burst so that we can estimate the background value of $\lambda$

$$\mathcal{I}(\Omega, t_b, t_e, m_b, m_e) = \int_{\mathcal{T}} \exp\left(\sum_{j=m_b}^{m_e} \left(\lambda_{(j+1)} - \lambda_{(j)}\right) x_{(j-m_b+1)}\right) dx \tag{8}$$

where $\mathcal{T} = \{x_1, \ldots, x_{m_e-m_b} : t_b \le x_1 \le x_2 \le \cdots \le x_{m_e-m_b} \le t_e\}$. For any $N$ and $a = (a_1, \ldots, a_N)$ define $G_a = G_a^N = \int_{\mathcal{T}} e^{\sum_{i=1}^N a_i x_i} dx$ where $\mathcal{T} = \{x_1, \ldots, x_N : T_1 \le x_1 \le x_2 \le \cdots \le x_{m_e-m_b} \le T_2\}$ (where we omit the dependence on $T_1, T_2$ for notational simplicity). Thus $\mathcal{I}(\Omega, t_b, t_e, m_b, m_e) = G_{\left(\lambda_{(j+1)}-\lambda_{(j)}\right)_{j \le m_e-m_b}}^{m_e-m_b}$ and computing $G_a$ for any choice of $a$ is enough.

First, we observe that we have the following recurrence relation

$$G_{a_1,\ldots,a_N}^N = \int_{T_1}^{T_2} \int_{x_1}^{T_2} \cdots \int_{x_{N-1}}^{T_2} e^{\sum_{i=1}^N a_i x_i} dx_N \ldots, dx_2 dx_1$$
$$= \int_{T_1}^{T_2} \int_{x_1}^{T_2} \cdots \int_{x_{N-2}}^{T_2} \left[\frac{e^{T_2 a_N} - e^{a_N x_{N-1}}}{a_N}\right] dx_{N-1} \ldots dx_2 dx_1$$
$$= \frac{e^{T_2 a_N} G_{a_1,\ldots,a_{N-1}}^{N-1}}{a_N} - \frac{e^{T_1 a_N} G_{a_1,\ldots,a_{N-1}+a_N}^{N-1}}{a_N}. \tag{9}$$

Based on iteratively applying this recurrence relation, we can get the following formula:

$$G_{a_1,\ldots,a_N}^N =$$
$$e^{T_1(a_1+\ldots+a_N)} \left( \sum_{\delta \in \{0,1\}^N} (-1)^{\sum_{i=1}^N \delta_i} \right.$$
$$\left. \times \left[ \frac{\prod_{\{i:\delta_i=0\}} \exp\left((T_2 - T_1)(a_i + \sum_{j=i+1}^N a_j \prod_{u=i+1}^j \delta_u)\right)}{\prod_{i=1}^N (a_i + \sum_{j=i+1}^N a_j \prod_{u=i+1}^j \delta_u)} \right] \right). \tag{10}$$

Whilst the iterative computation in question is reasonably straightforward, we reproduce the details here for the reader's convenience.

PROOF OF FORMULA (10). The proof is by induction. Note that for $N = 1$, we have indeed $G_{a_1}^N = \frac{e^{T_2 a_1}}{a_1} - \frac{e^{T_1 a_1}}{a_1}$ as expected. Suppose the result holds for $N - 1$ and let us prove it holds for $N$.

Note that by the scaling of the formula and the definition of $G$, it is clear that we can restrict ourselves to the case $T_1 = 0$. Now, by

equation (9), we have

$$G^N_{a_1,\ldots,a_N} = \frac{e^{T_2 a_N} G^{N-1}_{a_1,\ldots,a_{N-1}}}{a_N} - \frac{G^{N-1}_{a_1,\ldots,a_{N-1}+a_N}}{a_N} \tag{11}$$

$$= \frac{e^{T_2 a_N}}{a_N} \sum_{\delta \in \{0,1\}^{N-1}} \left( (-1)^{\sum_{i=1}^{N-1} \delta_i} \right.$$

$$\left. \times \left[ \frac{\prod_{i:\delta_i=0} \exp\left(T_2(a_i + \sum_{j=i+1}^{N-1} a_j \prod_{u=i+1}^{j} \delta_u)\right)}{\prod_{i=1}^{N-1}(a_i + \sum_{j=i+1}^{N-1} a_j \prod_{u=i+1}^{j} \delta_u)} \right] \right)$$

$$- \frac{1}{a_N} \sum_{\delta \in \{0,1\}^{N-1}} \left( (-1)^{\sum_{i=1}^{N-1} \delta_i} \right.$$

$$\left. \times \left[ \frac{\prod_{i:\delta_i=0} \exp\left(T_2(\tilde{a}_i + \sum_{j=i+1}^{N-1} \tilde{a}_j \prod_{u=i+1}^{j} \delta_u)\right)}{\prod_{i=1}^{N-1}(\tilde{a}_i + \sum_{j=i+1}^{N-1} \tilde{a}_j \prod_{u=i+1}^{j} \delta_u)} \right] \right), \tag{12}$$

where $\tilde{a}_i = a_i$ for $i \le N-2$ and $\tilde{a}_{N-1} = a_{N-1} + a_N$. Now, note that

$$-\frac{1}{a_n} \sum_{\delta \in \{0,1\}^{N-1}} (-1)^{\sum_{i=1}^{N-1} \delta_i} \left[ \frac{\prod_{i:\delta_i=0} \exp\left(T_2(\tilde{a}_i + \sum_{j=i+1}^{N-1} \tilde{a}_j \prod_{u=i+1}^{j} \delta_u)\right)}{\prod_{i=1}^{N-1}(\tilde{a}_i + \sum_{j=i+1}^{N-1} \tilde{a}_j \prod_{u=i+1}^{j} \delta_u)} \right] \tag{13}$$

$$= \frac{1}{a_n} \sum_{\delta \in \{0,1\}^{N-1} \times \{1\}} \left( (-1)^{\sum_{i=1}^{N} \delta_i} \right.$$

$$\left. \times \left[ \frac{\prod_{i:\delta_i=0} \exp\left(T_2(a_i + \sum_{j=i+1}^{N} a_j \prod_{u=i+1}^{j} \delta_u)\right)}{\prod_{i=1}^{N-1}(a_i + \sum_{j=i+1}^{N} a_j \prod_{u=i+1}^{j} \delta_u)} \right] \right)$$

$$= \sum_{\delta \in \{0,1\}^{N-1} \times \{1\}} \left( (-1)^{\sum_{i=1}^{N} \delta_i} \right.$$

$$\left. \times \left[ \frac{\prod_{i:\delta_i=0} \exp\left(T_2(a_i + \sum_{j=i+1}^{N} a_j \prod_{u=i+1}^{j} \delta_u)\right)}{\prod_{i=1}^{N}(a_i + \sum_{j=i+1}^{N} a_j \prod_{u=i+1}^{j} \delta_u)} \right] \right). \tag{14}$$

and

$$\frac{e^{T_2 a_N}}{a_N} \sum_{\delta \in \{0,1\}^{N-1}} (-1)^{\sum_{i=1}^{N-1} \delta_i} \left[ \frac{\prod_{\{i:\delta_i=0\}} \exp\left(T_2(a_i + \sum_{j=i+1}^{N-1} a_j \prod_{u=i+1}^{j} \delta_u)\right)}{\prod_{i=1}^{N-1}(a_i + \sum_{j=i+1}^{N-1} a_j \prod_{u=i+1}^{j} \delta_u)} \right] \tag{15}$$

$$= \frac{1}{a_N} \sum_{\delta \in \{0,1\}^{N-1} \times \{0\}} (-1)^{\sum_{i=1}^{N} \delta_i} \left[ \frac{\prod_{\{i:\delta_i=0\}} \exp\left(T_2(a_i + \sum_{j=i+1}^{N} a_j \prod_{u=i+1}^{j} \delta_u)\right)}{\prod_{i=1}^{N-1}(a_i + \sum_{j=i+1}^{N-1} a_j \prod_{u=i+1}^{j} \delta_u)} \right]$$

$$= \sum_{\delta \in \{0,1\}^{N-1} \times \{0\}} (-1)^{\sum_{i=1}^{N} \delta_i} \left[ \frac{\prod_{i:\delta_i=0} \exp\left(T_2(a_i + \sum_{j=i+1}^{N} a_j \prod_{u=i+1}^{j} \delta_u)\right)}{\prod_{i=1}^{N}(a_i + \sum_{j=i+1}^{N} a_j \prod_{u=i+1}^{j} \delta_u)} \right]. \tag{16}$$

Plugging equations (13) and (15) back into equation (11), we obtain

$$G^N_{a_1,\ldots,a_N} = \frac{e^{T_2 a_N}}{a_N} \sum_{\delta \in \{0,1\}^{N-1}} \left( (-1)^{\sum_{i=1}^{N-1} \delta_i} \right.$$

$$\left. \times \left[ \frac{\prod_{i:\delta_i=0} \exp\left(T_2(a_i + \sum_{j=i+1}^{N-1} a_j \prod_{u=i+1}^{j} \delta_u)\right)}{\prod_{i=1}^{N-1}(a_i + \sum_{j=i+1}^{N-1} a_j \prod_{u=i+1}^{j} \delta_u)} \right] \right)$$

$$- \frac{1}{a_N} \sum_{\delta \in \{0,1\}^{N-1}} \left( (-1)^{\sum_{i=1}^{N-1} \delta_i} \right.$$

$$\left. \times \left[ \frac{\prod_{i:\delta_i=0} \exp\left(T_2(\tilde{a}_i + \sum_{j=i+1}^{N-1} \tilde{a}_j \prod_{u=i+1}^{j} \delta_u)\right)}{\prod_{i=1}^{N-1}(\tilde{a}_i + \sum_{j=i+1}^{N-1} \tilde{a}_j \prod_{u=i+1}^{j} \delta_u)} \right] \right)$$

$$= \sum_{\delta \in \{0,1\}^{N}} \left( (-1)^{\sum_{i=1}^{N} \delta_i} \right.$$

$$\left. \times \left[ \frac{\prod_{i:\delta_i=0} \exp\left(T_2(a_i + \sum_{j=i+1}^{N} a_j \prod_{u=i+1}^{j} \delta_u)\right)}{\prod_{i=1}^{N}(a_i + \sum_{j=i+1}^{N} a_j \prod_{u=i+1}^{j} \delta_u)} \right] \right). \tag{17}$$

$\square$

# D TABLE OF NOTATIONS

In this section, we provide a table of notations used in this paper. There are a small number of duplicates to aid understanding when the same notation is relevant to different parts of the paper.

| Notation | Meaning |
| --- | --- |
| **In the definition of our model process** | |
| $t_1, t_2, \ldots$ | Event timestamps |
| $T = \{t_1, t_2, \ldots\}$ | Set of event timestamps |
| $\Delta t_i$ | $t_i - t_{i-1}$ |
| $(\mathcal{H}_t)_{t \in \mathbb{R}}$ | Filtration |
| $\mathcal{H}_t$ | Information realised at time $t$ |
| SFP | Self-feeding Process |
| NHPP | Non-homogeneous Poisson Process |
| MPP | MetaPoisson Process (latent transition process) |
| $z_i$ | Label of $i$th timestamp |
| $z_i = 1$ if | $t_i \in$ SFP |
| $z_i = 0$ if | $t_i \in$ NHPP |
| $\lambda_s(T\|\mathcal{H}_t)$ | Intensity of the Self-feeding Process (SFP) |
| | $\lambda_s(T\|\mathcal{H}_t) = 1/(\mu/e + g(t) - g(g(t)))$ |
| $g(t) = [\max(t_i : z_i = 1 \quad \wedge \quad t_i < u)]_+$ | Timestamp of last SFP process |
| $\Phi = \{\varphi_1, \varphi_2, \ldots\} \subset \mathbb{R}^+$ | Unobserved 'Metapoisson' (MPP) events |
| $\lambda_\phi(t) = c\lambda_s(t)$ | Intensity of the latent transition (MPP) |
| $c \in [0,1]$ | Parameter that controls the NHPP transition sensitivity |
| $\lambda_p(t) = \lambda_{m_t}$ | Intensity of the Non Homogeneous Poisson Process (NHPP) |
| $m_t = \sum_{j=1}^{n_\phi} 1_{\varphi_j < t}$ | Number of Metapoisson transitions occurred so far |
| $M := \{m_1, m_2, \cdots, m_n\}$ | set of Metapoisson transitions |
| | = index of current NHPP intensity |
| $\lambda_s^+(t) = \lim_{\delta \to 0} \lambda_s(t + \delta)$ | Intensity of SPF immediately after time $s$ |
| $\lambda_\phi^+(t)$ | Intensity of MetaPoisson immediately after time $s$ |
| $\lambda_p^+(t)$ | Intensity of NHPP immediately after time $s$ |
| **In the detailed explanation of the generation procedure** | |
| $E_s$ | Exponential random variable with parameter $\lambda_s^+(t)$ |
| $E_\phi$ | Exponential random variable with parameter $\lambda_\phi^+(t)$ |
| $E_p$ | Exponential random variable with parameter $\lambda_p^+(t)$ |
| $t + E := t + \min(E_s, E_\phi, E_p)$ | Timestamp of next event or latent transition |
| **In the explanation of our EM-algorithm (E-Step)** | |
| $N(t) = \sum_{i=1}^n 1_{t_i \le t}$ | Cumulative number of observed events up to time $t$ |
| $\Phi_i := \{\varphi_j \| t_{i-1} < \varphi_j < t_{i+1}\}$ | Set of NHPP transitions between $t_{i-1}$ and $t_{i+1}$ |
| $M := \{m_1, m_2, \cdots, m_n\}$ | Set of Metapoisson transitions |
| $\theta = Z \cup \Phi \cup M$ | Set of all *latent* variables |
| $\theta_i := (\{z_i, m_i\} \cup \Phi_i)$ | Latent variables associated to time $t_i$ |
| $\theta_{-i} = \theta \setminus \theta_i$ | Latent variables except those around $t_i$ |
| $N_{\text{Gibbs}}$ | Number of component updates |
| | in one iteration of Gibbs algorithm |
| $\mathcal{L}(\theta) =$ | |
| $\prod_{i=1}^n \lambda_s(t_i)^{z_i} \lambda_p(t_i)^{1-z_i} \prod_{j=1}^{n_\phi} \lambda_\phi(\varphi_j) \times e^{-\int_0^{t_n} \lambda_s(t) + \lambda_\phi(t) + \lambda_p(t)dt}$ | Likelihood of our model |
| $\mathcal{L}(\theta) = \mathcal{L}_s(\theta)\mathcal{L}_\phi(\theta)\mathcal{L}_p(\theta)$ | Factorized form of likelihood |
| $\mathcal{L}_s(\theta) = \prod_{i=1}^n \lambda_s(t_i)^{z_i} e^{-\int_0^{t_n} \lambda_s(t)dt}$ | SFP component of likelihood |
| $\mathcal{L}_\phi(\theta) = \prod_{j=1}^{n_\phi} \lambda_\phi(\varphi_j) \times e^{-\int_0^{t_n} \lambda_\phi(t)dt}$ | Metapoisson component of likelihood |
| $\mathcal{L}_p(\theta) = \prod_{i=1}^n \lambda_p(t_i)^{1-z_i} \times e^{-\int_0^{t_n} \lambda_p(t)dt}$ | NHPP component of likelihood |
| $f(u) = \text{argmin}_j \{t_j \| t_u < t_j \wedge z_j = 1\}$ | index of the next SFP event after $t_u$ |
| $\Omega_{z,m} := \theta_{-i} \cup \{z, m\}$ | All latent variables excluding $\Phi_i$ |
| | $\theta = \Omega_{z,m} \cup \Phi_i$ |

| | |
|---|---|
| $\mathbb{P}(z_i = z, m_i = m \| T, \theta_{-i}) \propto \mathcal{L}_s^i(\Omega_{z,m}) \mathcal{L}_\phi^i(\Omega_{z,m}) \mathcal{L}_p^i(\Omega_{z,m})$ | conditional distribution of $(z_i, m_i)$ given all variables except $\theta_{-i}$ |
| $\mathcal{L}_s^i(\Omega_{z,m})$ | component of the **SFP** likelihood in the interval $[t_{i-1}, t_{f(f(i))})$ corresponding to $z_i = z$, $m_i = m$ |
| $\mathcal{L}_\phi^i(\Omega_{z,m})$ | component of the **MPP** likelihood in the interval $[t_{i-1}, t_{f(f(i))})$ corresponding to $z_i = z$, $m_i = m$ |
| $\mathcal{L}_p^i(\Omega_{z,m}) := \int_{\Phi_i \in F_{\Omega_{z,m}}} \mathcal{L}_p^i(\Omega_{z,m} \cup \Phi_i) d\Phi_i$ | component of the **NHPP** likelihood in the interval $[t_{i-1}, t_{f(f(i))})$ corresponding to $z_i = z$, $m_i = m$ |
| $F_{\Omega_{z,m}}$ | Set of $\Phi_i$s compatible with the values of $T, M$ when $m_i$ is set to the index $m$ (cf. explanation below equation (3)) |
| $C$ | $e^{t_{i-1}\lambda_p(t_{t-i})} e^{-t_{i+1}\lambda_p(t_{t+i})} e^{-\int_{t_{i+1}}^{t_{f(f(i))}} \lambda_p(t)dt}$ (normalization constant in expression of $\mathcal{L}_p^i(\Omega_{z,m})$, cf. equation (4)) |
| $\mathcal{T} = \{x = (x_1, \ldots, x_{m_e - m_b}) \| t_b \le x_1 \le x_2 \le \cdots \le x_{m_e - m_b} \le t_e\}$ | Domain of possible MPP incremental positions between $t_b$ and $t_e$ assuming $m_b$ MPP events at $t_b$ and $m_e$ at $t_e$ |
| $\mathcal{I}(\Omega, t_b, t_e, m_b, m_e)$ | $\int_{\mathcal{T}} e^{\sum_{j=m_b}^{m_e} (\lambda_{(j+1)} - \lambda_{(j)}) x_{(j-m_b+1)}} dx$ |
| $\Phi_i^- := \{\varphi_j \| t_{i-1} < \varphi_j < t_i\}$ | Set of MPP transitions between $t_{i-1}$ and $t_i$ |
| $\Phi_i^+ := \{\varphi_j \| t_i < \varphi_j < t_{i+1}\}$ | Set of MPP transitions between $t_i$ and $t_{i+1}$ |
| $N_{a,b} = \#(j : \varphi_j \in [a, b])$ | Number of MPP between $a$ and $b$ |
| **In the explanation of our EM-algorithm (M-Step)** | |
| $N_\theta$ | Number of samples of $\theta$ drawn from conditional distribution of $\theta$ given the current estimate of $\{\mu, \Lambda\}$ |
| $\{\theta_1, \theta_2, \cdots, \theta_{N_\theta}\}$ | Samples of $\theta$ |
| $\hat{\mu} = \left(\sum_{j=1}^{N_\theta} \arg\min_\mu \log(\mathcal{L}_s(\theta_j))\right) / N_\theta$, | Update for $\mu$ |
| $\bar{\Phi}$ | Ordered list of the elements of $\{0 \cup \Phi \cup t_n\}$ |
| Thus: | $\bar{\varphi}_0 = 0$, and $\bar{\varphi}_{m_t + 1} = t_n$ |
| $U_j(\theta) = \sum_j 1_{t_j \in [\bar{\varphi}_j, \bar{\varphi}_{j+1}) \wedge z_j = 0}$ | Number of observed timestamps between $j$th and $j+1$th (observed and non observed) events |
| **In the warm start procedure** | |
| $I_i = [(i-1)t_n/n_k, it_n/n_k)$ | $i$th sub interval in the split of $[0, t_n)$ |
| $u_i = \sum_{j \le n} 1_{t_j \in I_i}$ | the number of events belonging to interval $I_i$ |
| $I^1, I^2, \ldots, I^k$ | intervals subsampled without replacement with $\mathbb{P}(I_i) \propto u_i$ |
| $k$ | hyperparameter |
| $\Phi$ | Set of transitions sampled uniformly from each of $I^1, I^2, \ldots, I^k$ |
| $I_0^j, I_1^j, \ldots, I_{n_k-1}^j$ | $n_k$ sub-intervals of equal length partitioning $[\bar{\varphi}_j, \bar{\varphi}_{j+1})$ |
| $u_i^j = \sum_{o \le n} 1_{t_o \in I_i^j}$ | Number of observed events in $I_i^j$ |
| $I_v^j$ | Interval subsampled with $\mathbb{P}(v = i) \propto 1/u_i^j$ (except when $u_i^j = 0$ in which case $\mathbb{P}(v = i) = 0$) |
| $\lambda_j = (u_{j_v} n_k)/(\bar{\varphi}_{j+1} - \bar{\varphi}_j)$ | Initial value of $\lambda_j$ after warm start |
| $p_i = \min(\lambda_{m_i}(\varphi_{m_i+1} - \varphi_{m_i})/\|\Delta_i\|, 1)$ | Probability parameter of Bernouilli used to generate warm start labels |
| $\Delta_i = \{t_l \| \bar{\varphi}_{m_i} < t_l < \bar{\varphi}_{m_i+1}\}$ | Number of (observed) events between $\bar{\varphi}_{m_i}$ and $\bar{\varphi}_{m_i+1}$ |
| $\mu = \text{median}(\{t_i \| z_i = 1\})$ | Initial value of $\mu$ after warm start |
| **In the calculation of the integral** | |
| $\mathcal{I}(\Omega, t_b, t_e, m_b, m_e)$ | $\int_{\mathcal{T}} e^{\sum_{j=m_b}^{m_e} (\lambda_{(j+1)} - \lambda_{(j)}) x_{(j-m_b+1)}} dx$ |
| $\mathcal{T}$ | $\{x_1, \ldots, x_N : T_1 \le x_1 \le x_2 \le \cdots \le x_{m_e - m_b} \le T_2\}$ |
| $G_a = G_a^N$ | $\int_{\mathcal{T}} e^{\sum_{i=1}^N a_i x_i} dx$ |
| $\mathcal{I}(\Omega, t_b, t_e, m_b, m_e) = G_{(\lambda_{(j+1)} - \lambda_{(j)})_{j \le m_e - m_b}}^{m_e - m_b}$ | Expression of target $\mathcal{I}$ as a function of the more general quantity $G_a$ |
| $\delta \in \{0, 1\}^n$ | Dummy variable for label assignments |
| **In the real data analysis** | |
| $[0, 1] \ni \kappa = \int_a^b \frac{\lambda_p(t)(b-a)^{-1}}{\lambda_p(t) + \lambda_s(t)} dt$ | Absolute stability |

| | |
|---|---|
| $[0,1] \ni \widetilde{\kappa} = \dfrac{\int_a^b \lambda_p(t)dt}{\int_a^b (\lambda_p(t)+\lambda_s(t))dt}$ | Relative stability |
| $\lambda(t\|\mathcal{H}_t) = \lambda + \sum_{t_i < t} K(t - t_i)$ | Hawkes process intensity function |
| $K$ | Kernel function |
| $R^2_{Hawkes}$ | Determination coefficient for Hawkes model |
| $R^2_S$ | Determination coefficient of **SFP** component of [18] |
| $R^2_{HPP}$ | Determination coefficient of **Homogeneous Poisson Process (HPP)** component of BuSca [18] |
| $R^2_{BP} = (R^2_S + R^2_{HPP})/2$ | Final determination coefficient of BuSca [18] |
| $R^2_{SFP}$ | determination coefficient of SFP component of **BPoP** |
| $R^2_{NHPP_i}$ | Determination coefficient for LR problem predicting $N(t)$ from $t$ on $[\bar{\varphi}_i, \bar{\varphi}_{i+1})$ |
| $R^2_{NHPP} = \sum_i (\bar{\varphi}_{i+1} - \bar{\varphi}_i) R^2_{NHPP_i}/t_n$ | Determination coefficient for NHPP component of **BPoP** |
| $R^2_{\mathbf{BPoP}} = (R^2_{SFP} + R^2_{NHPP})/2$ | Overall determination coefficient of **BPoP** |
| **In the synthetic data generation** | |
| $P_{NHPP} = \sum_{i=1}^n (1 - z_i)/n$ | Proportion of observed events that belong to NHPP |
| $(\mathbb{E}(n), \mathbb{E}(\|\Phi\|), \mathbb{E}(P_{NHPP}))$ | Quantities of interest to pick interesting configurations of hyperparameters $\mu$, $\Lambda$ and $c$ |
| $\delta(\theta, \hat{\theta}) = \int_0^{t_n} \dfrac{\|(\lambda_s(t\|\theta)+\lambda_p(t\|\theta)) - (\lambda_s(t\|\hat{\theta})+\lambda_p(t\|\hat{\theta}))\|}{\lambda_s(t\|\theta)+\lambda_p(t\|\theta)} dt.$ | Performance measure (weighted accuracy of recovered labelling) |

