# OpenReview forum: "Unraveling the Dynamics of Stable and Curious Audiences in Web Systems"
_ACM.org/TheWebConf/2024/Conference — TheWebConf24 Oral_

### Official Review · Reviewer_wJeT · 2023-11-10

**Novelty:** 4
**Technical Quality:** 4

**Review:**

This paper presents a model designed for distinguishing stable and curious media audiences which is an important task with a wide range of applications. The proposed method, BPoP, employs a random series of events, without any external information, to disentangle these two types of audiences. Their parameters are optimized by the proposed EM algorithm, effectively addressing the complexity inherent in the problem. The experimental findings demonstrate BPoP's effectiveness in distinguishing stable and ephemeral audiences in real-world event series datasets.

However, the paper's methodology is hard to follow due to its numerous notations. To enhance clarity and readability, incorporating a table summarizing the meaning of each notation is recommended. In addition, the experimental results lack numerical evaluation due to the absence of ground-truth labels in the datasets.

**Questions:**

- Please find additional concerns above.
- Could the authors elucidate why the observation that 91% of individuals exhibit a value of 0.05 < P_NHPP < 0.96 implies a combination of stable and curious audience?
- Are there any existing deep learning-based methods that address the same or similar problem?

**Reviewer Confidence:**

2: The reviewer is willing to defend the evaluation, but it is likely that the reviewer did not understand parts of the paper

**Scope:**

4: The work is relevant to the Web and to the track, and is of broad interest to the community

---

### Official Review · Reviewer_ShZH · 2023-11-17

**Novelty:** 6
**Technical Quality:** 6

**Review:**

PC members of TheWebConf in previous editions were asked to bid for papers to review. This allowed reviewers to volunteer for articles that were strongly connected to their research interests and background. In this edition, the assignment of papers to review has been done automatically by a system that matches the content of the submissions and the content of past papers of the reviewers. As a result, it may happen that a reviewer receives an article for which they have limited background for a thorough review. This is my situation with this submission: I am able to evaluate the main ideas of this work but not the core and details of the modeling approach, so my review below will be somewhat superficial.

This submission introduces a model to analyze social media activity time series. The idea behind the model is to combine two different processes: activity from both stable and curious audiences. The approach is validated empirically with real and synthetic data providing multiple findings.

I personally appreciate the narrative of the manuscript clearly explaining from the introduction the motivation to combine the modeling of consistent activity by stable audiences and the bursty activity of curious audiences. The review of related work allows the authors to make claims on the limitations of past approaches based only on Poisson, Hawkes and Wold processes. I can think of additional literature that could have been cited (e.g, [1,2]), however, I find the related work section comprehensive.

I also give special value to the work done by the authors in evaluating their model with up to 11  datasets of activity from web platforms of a very different nature. I am particularly curious that in the case of Twitter, the example of the hashtag #ACL has been chosen. While there are users who used this hashtag for posts about the Austin City Limits event, it is also used for other events such as the Asian Champions League or even the Association for Computational Linguistics. In that sense, one would wonder what is the noise level of the real-world datasets used for evaluating the model.

Despite the above issue, the modeling approach and the analysis of the findings from the 11 datasets  are likely to be of interest to many members of the TheWebConf community, making this submission a very relevant work to be considered.

# References
1. Fujita, K., Medvedev, A., Koyama, S., Lambiotte, R., & Shinomoto, S. (2018). Identifying exogenous and endogenous activity in social media. Physical Review E, 98(5), 052304.
2. Yin, H., Cui, B., Lu, H., Huang, Y., & Yao, J. (2013, April). A unified model for stable and temporal topic detection from social media data. In 2013 IEEE 29th International Conference on Data Engineering (ICDE) (pp. 661-672). IEEE.

**Questions:**

My only question relates to the issue mentioned in the review about the empirical data: could you provide a more extensive description of how all these data sets were generated?

**Ethics Review Description:**

-

**Reviewer Confidence:**

2: The reviewer is willing to defend the evaluation, but it is likely that the reviewer did not understand parts of the paper

**Scope:**

4: The work is relevant to the Web and to the track, and is of broad interest to the community

---

### Official Review · Reviewer_U4Mv · 2023-11-23

**Novelty:** 6
**Technical Quality:** 6

**Review:**

The authors propose a method called BPoP (Burst-induced Poisson Process), which is designed to decouple time series data (usually events on web) into two components---one models the stable audience who have sustained interests and the other models the curious audience who are drawn by sporadic bursty events. To achieve this, BPoP has three components: a non-homogeneous Poisson process (NHPP), a self-feeding process (SFP), and a meta Poisson process (MPP). Based on the authors, the key novelty is the introduction of MPP, which allows the NHPP to transit (i.e., switching to a new NHPP). The intuition of MPP is that stable audience may change their behaviors when a significant bursty event happens. This is nicely illustrated by Figure 1, the sustained interests of Michael Jackson increase after his decease, and the sustained interests of Barack Obama decrease after his presidential term concludes. The authors tested and evaluated BPoP on an extensive set of datasets over different platforms and various kinds of events. They reported superior performance of BPoP compared to a vanilla Hawkes process and BuSca (SFP + NHPP). The authors also derived two metrics: absolute stability and relative stability. They presented several analyses based on the two metrics.

**Pros:**
The design of BPoP is well motivated and experiments are comprehensive and solid. The accompanied analyses are easy to understand and demonstrated with lots of intuitive examples.

**Cons:**
While this is a solid submission, I do find the novelty a bit weak, because there is a rich line of prior work exploring SFP + NHPP, SFP + MPP, and NHPP + MPP.

**Rebuttal:**
I have read the rebuttal and updated the score.

**Questions:**

1. I think the authors missed the line of state-dependent Hawkes processes (i.e., SFP + MPP). For example, [1] and [2]. The core idea is similar in a way that all allow the underlying model to have state transitions.

* [1] State-dependent Hawkes processes and their application to limit order book modelling. https://arxiv.org/abs/1809.08060
* [2] Nonlinear Hawkes Processes in Time-Varying System. https://arxiv.org/abs/2106.04844

2. I am confused with the data in those 11 real-world datasets. Most of the datasets contains individual events and their corresponding timestamps, for example, a single comment and its timestamp. But I am not sure about Github (Users), Github (Projects), and Google Trends. I am less familiar with Github data, but at least for Google Trends, I believe Google Trends returns a time series of normalized interests, and the timestamp is either a day or a week. I tried Google Trends on Michael Jackson from Jan 01 2008 to Dec 31 2010.  Each data point is a week (e.g., Dec 7-13, 2008) along with the relative normalized interest in that week. So this Google Trends time series does not indicate a single event, but aggregate volume of events (herein an event is a search of Michael Jackson on YouTube) over a week. I am not sure how the authors handled such data, there are at least two questions:

* The time interval $\Delta t_i$ is a fixed value now (e.g., a week) in this case, how did you model that in BPoP?
* How did you model the weight of each event? In the Google Trends example, the weight is the value of relative normalized interests.

3. In Figure 1, the x axis %t and y axis %n(t) are not explained. They are not in the Appendix D table of notations, either. I guess %n(t) is the RSE you observed. I suggest replacing %t with calendar datetime and adding annotation for the vertical black line. I think it will also be good to list the derived metrics here, e.g., $\kappa$, $\tilde \kappa$, $\Phi$, and refer to the later section for the calculation of those metrics. Another question: is $\Phi = 2$ for both "Michael Jackson" and "Barack Obama" RSE?

4. Figure 3 caption, should B be $t_i \notin NHPP$?

5. I still have difficulties understanding the difference between $\kappa$, $\tilde \kappa$, and $\Phi$. I think $\tilde \kappa$ is straightforward--- the proportion of the activity in the interval $[𝑎, 𝑏]$ that is assigned to the stable audience. So $\tilde \kappa$ should be bounded between 0 and 1. But what is $\kappa$? What is the importance of the stable audience averaged over the time interval $[𝑎, 𝑏]$? How to understand this importance? Can you give a laymen's term for $\kappa$ and $\Phi$?

6. In Figure 4, black vertical lines need explanations. I am just guessing, is the $\Phi = 0$ for A. Claire, Redfoo, and Stacey Q while $\Phi = 1$ for B. Mendler and B. Eilish?

7. Section 4 **Datasets** paragraph. Some datasets are taken from prior literature with footnotes, but there are a few without any details about the data collection. I suggest adding a section in the Appendix, and describing the data collection processes. If the data is from a prior work, then describe it briefly; if the data was collected by the authors, then describe it in detail.

**Reviewer Confidence:**

3: The reviewer is confident but not certain that the evaluation is correct

**Scope:**

4: The work is relevant to the Web and to the track, and is of broad interest to the community

---

### Official Review · Reviewer_ZSwE · 2023-11-24

**Novelty:** 5
**Technical Quality:** 5

**Review:**

The paper proposes a Burst-induced Poisson Process (BPoP) model that can effectively separate the slowly-varying regular activity of the stable audience from the bursty behavior caused by curious audiences. The author shows that BPoP efficiently captures the time-varying background rates that realistically represent the stable audience.

​	   Strengths:

1. This paper provides a fresh perspective on time series, which provides a clear distinction between different types of audience engagement and contributes to a more accurate analysis of time series data

2. The proposed method is evaluated in multiple datasets and shows its superiority in performance.

3. The paper is well-written and easily understandable.

   Weaknesses:

1. Methodology:  The paper lacks     further explanation on how the self-feeding process (SFP) module is able to generate all types of bursty events and effectively differentiate them     from normal events. Providing more details on this aspect could enhance the understanding of the model's capabilities.

2. Experiment: The evaluation of the proposed model could be strengthened by including more baseline models.

3. Application: The paper does not  thoroughly explain how the popularity of time series can be evaluated using the proposed BPoP model. It would be beneficial to provide a     detailed example that demonstrates the specific analysis and evaluation process.

**Questions:**

refer to the above questions

**Reviewer Confidence:**

3: The reviewer is confident but not certain that the evaluation is correct

**Scope:**

4: The work is relevant to the Web and to the track, and is of broad interest to the community

---

### Official Review · Reviewer_P57q · 2023-11-25

**Novelty:** 4
**Technical Quality:** 4

**Review:**

This paper proposes burst-induced Poisson process for analysis of timeseries, then it fits the proposed model to several time series data-sets. The idea of the model is a try to split the increase of the serries, e.g., online audiences , into curious audience - joining due to some event, or stable - that join some channel  staidly. I think that the paper has some merit, but in my opinion it misses the ablation study. The current narrative of the paper is: we build more complex model -> we do experiments to show that it fits to the data better. This is highly expected and from such experiment we do not learn any phenomena nor learn any explanations about what happens. Hence, an extension o the paper with a section that would more carefully look on the impact of different assumption of the model on the fit is expected.

**Questions:**

Please try to explain what did we learn from the experiment you have provided? And in what way the results/model you have proposed can be useful in practice?

**Ethics Review Description:**

Not relevant.

**Reviewer Confidence:**

3: The reviewer is confident but not certain that the evaluation is correct

**Scope:**

3: The work is somewhat relevant to the Web and to the track, and is of narrow interest to a sub-community

---

### Decision · Program_Chairs · 2024-01-22

**Decision:**

Accept (Oral)

**Comment:**

This paper proposes a model, called Burst-induced Poisson Process [BPoP], of those who engage with web systems, that consists of a mixture of an ongoing "stable" (though potentially slowly varying) audience accompanied by a "curious" audience that's burstier in its attention. The paper compares the model to several real-world time-series datasets and existing models in the literature.

 The reviewers appreciated the novel mixture-of-audiences perspective on time series and the empirical evaluation. Most also praise the writing, organization, and readability of the paper (though some found the notation a bit excessive, particularly wanting a concise table of notation in the paper proper rather than in the appendix).

 The weaknesses identified by the reviewers include a lack of attention to whether the added complexity of the proposed model is justified by its improved results, and some sense of a lack of comparison/novelty relative to existing work in the literature. [Note: the bibliography sorted by citation order rather than author name is hard to use.] While not all reviewers were able to respond to the authors' replies, those who were generally found the proposed revisions that the authors mentioned to substantively address at least some of their concerns.